# Cis- and trans-regulations of pre-mRNA splicing by RNA editing enzymes influence cancer development

Sze Jing Tang [1,4], Haoqing Shen [1,2,4], Omer An [1], HuiQi Hong[1,3], Jia Li[1], Yangyang Song [1], Jian Han [1], Daryl Jin Tai Tay [1], Vanessa Hui En Ng [1], Fernando Bellido Molias [1], Ka Wai Leong [1], Priyankaa Pitcheshwar [1,2], Henry Yang[1] & Leilei Chen [1,2]*

RNA editing and splicing are the two major processes that dynamically regulate human transcriptome diversity. Despite growing evidence of crosstalk between RNA editing enzymes (mainly ADAR1) and splicing machineries, detailed mechanistic explanations and their biological importance in diseases, such as cancer are still lacking. Herein, we identify approximately a hundred high-confidence splicing events altered by ADAR1 and/or ADAR2, and ADAR1 or ADAR2 protein can regulate cassette exons in both directions. We unravel a binding tendency of ADARs to dsRNAs that involves GA-rich sequences for editing and splicing regulation. ADAR1 edits an intronic splicing silencer, leading to recruitment of SRSF7 and repression of exon inclusion. We also present a mechanism through which ADAR2 binds to dsRNA formed between GA-rich sequences and polypyrimidine (Py)-tract and precludes access of U2AF65 to 3′ splice site. Furthermore, we find these ADARs-regulated splicing changes per se influence tumorigenesis, not merely byproducts of ADARs editing and binding.

[1] Cancer Science Institute of Singapore, National University of Singapore, Singapore 117599, Singapore. [2] Department of Anatomy, Yong Loo Lin School of Medicine, National University of Singapore, Singapore 117594, Singapore. [3] Department of Physiology, Yong Loo Lin School of Medicine, National University of Singapore, Singapore 117549, Singapore. [4] These authors contributed equally: Sze Jing Tang, Haoqing Shen. *email: polly_chen@nus.edu.sg

Alternative splicing and RNA editing are the two major pre-messenger RNA (pre-mRNA) processing steps diversifying transcriptome in eukaryotes. Regulation of alternative splicing involves *cis*-acting elements, such as exonic/intronic splicing enhancers/silencers (ESE/S, ISE/S), and *trans*-acting factors, such as serine arginine-rich (SR) proteins and heterogeneous nuclear ribonucleoproteins (hnRNPs), transcription, chromatin modification, and RNA secondary structure[1–11]. Aberrant splicing is associated with numerous cancers and it is generally caused by mutations in *cis*- and *trans*-splicing regulatory elements and altered expression of splicing factors[4]. RNA editing can introduce changes in RNA sequences encoded by the genome, contributing to RNA mutations. Adenosine deaminases acting on RNA (ADARs) family of enzymes, ADAR1 and ADAR2 (ADAR1 and 2), catalyze editing of adenosine to inosine (A-to-I) in double-stranded RNA (dsRNA), which is the most common type of RNA editing in mammals[12,13]. Due to structural similarity, inosine is recognized as guanosine by translation and splicing machineries, resulting in protein recoding and splicing change, respectively[13–17]. Dysregulated ADAR1 and 2 expression, and RNA editing are also implicated in neurological disorders and cancer[18].

It is expected that RNA editing and splicing have extensive crosstalk, as they share the largest overlap in time and space during processing of pre-mRNA[19]. Editing occurs co-transcriptionally on nascent RNA, suggesting that editing precedes splicing and may cause widespread effects on splicing[20]. As proposed by previous studies, A-to-I editing may modulate splicing through disrupting branch point sequence (BPS)[21], creating novel 3′ splice site (3′ss)[17,20,22], modifying auxiliary *cis*-acting elements[23,24] as well as affecting RNA secondary structure[25,26]. However, the majority of studies were lack of experimental validations, specific mechanisms, and functional importance of crosstalk events. Since very few canonical *cis*-acting elements (5′ splice site (5′ss), 3′ss, and BPS) are affected by editing[27], we believe that ADAR proteins may regulate splicing through other mechanisms beyond their editing functions[20]. Albeit both ADAR1 and 2 edit many sites, there is very little information about the role of ADAR2 in splicing regulation. It also remains poorly understood whether ADAR1 and 2 exert their regulations on splicing through the same or distinct mechanisms. In addition, functional importance of splicing events regulated by ADARs in human diseases, including cancer remains largely unknown.

In this study, we seek to investigate the crosstalk between RNA editing and splicing machineries in the context of cancer, as dysregulations of both machineries are implicated in cancer[14,16,28–38]. Approximately a hundred high-confidence splicing events regulated by ADAR1 or/and 2 are identified through a comprehensive transcriptome-wide RNA-sequencing (RNA-Seq). Two cassette exon targets, *CCDC15* (coiled-coil domain containing 15) exon 9 (*CCDC15*-ex9) and *RELL2* (receptor expressed in lymphoid tissues-like 2) exon 3 (*RELL2*-ex3) are chosen as exemplary targets to decipher the interplay between editing enzymes and splicing machineries, and their biological importance in cancer.

## Results

**ADARs regulate splicing dependent/independent of editing.** We modulated expression of ADAR1/2 in an esophageal squamous carcinoma (ESCC) cell line EC109 that has been used for RNA editing studies previously[33,34], using lentivirus-mediated silencing and overexpression (Fig. 1a), followed by transcriptome-wide RNA-Seq analysis of editing sites and alternative splicing events. A-to-I RNA editome was dramatically activated or repressed by overexpression or silencing of ADAR1/2, respectively (Supplementary Fig. 1a). Approximately a hundred high-confidence splicing events (cassette exon, intron retention, and alternative 5′ or 3′ss) regulated by ADAR1 or/and 2 were identified by fulfilling stringent requirements (total junction reads ≥ 15, splicing index change ($|\Delta SI|$) ≥ 10% with a false discovery rate (FDR) adjust *p*-value < 0.2; and splicing index (SI) change from both knockdown and overexpression experiments must be in opposite direction; Fig. 1b). Among the affected cassette exon events, exon inclusion could be either repressed or promoted by ADAR proteins (Fig. 1c). Selected targets could be experimentally validated in both original RNA-Seq EC109 cells and HEK293T cells (Fig. 1d, e, Supplementary Fig. 1b). We then selected *CCDC15*-ex9 and *RELL2*-ex3 as exemplary crosstalk targets for further investigations.

To understand whether ADARs regulate *CCDC15*-ex9 and *RELL2*-ex3 inclusion through or beyond their editing functions, we generated different ADAR1/2 mutants devoid of either enzymatic activity (DeAD mutants)[39] or dsRNA-binding capability (EAA mutants)[40,41]. In both HEK293T and EC109 cells, *CCDC15*-ex9 inclusion was repressed by wild-type ADAR1; intriguingly, it was also repressed by wild-type ADAR2 (Fig. 2a, Supplementary Fig. 2a), which failed to be identified by RNA-Seq due to insufficient junction reads detected in ADAR2 RNA-Seq samples. To confirm this observation, we overexpressed ADAR1/2 in HEK293T cells and found that inclusion of *CCDC15*-ex9 decreased dose dependently (Fig. 2b and Supplementary Fig. 2b), suggesting that this inclusion was indeed repressed by both ADAR proteins. Of note, DeAD mutant of ADAR1 retained a subtle impact on *CCDC15*-ex9 inclusion; while DeAD mutant of ADAR2 caused the same effect as its wild-type, and both of EAA mutants were incapable of repressing the inclusion (Fig. 2a). All these data indicated that editing and dsRNA-binding functions of ADAR1 are essential for its repression of *CCDC15*-ex9 inclusion; while for ADAR2, dsRNA-binding but not editing is indispensable for splicing regulation. On the other hand, *RELL2*-ex3 inclusion was repressed by both ADAR2 wild-type and DeAD mutant, but not EAA mutant (Fig. 2c), suggesting that the regulation requires dsRNA binding but not editing function of ADAR2. Altogether, ADAR proteins could regulate exon inclusion through both editing-dependent and -independent mechanisms.

**ADAR1-mediated editing represses *CCDC15* exon 9 inclusion.** To investigate whether ADAR1-mediated editing indeed affects *CCDC15* splicing, we first searched for editing sites in exon 9 and flanking introns. We identified three ADAR1-regulated editing sites (sites 1, 2, and 4) and an ADAR2-specific editing site (site 3) at a GA-rich hotspot region 240-nt upstream of the intron 8–exon 9 junction (Fig. 3a). We subsequently generated a minigene consisting of *CCDC15* exons 8–10 and intervening introns, and introduced an A-to-G point mutation to the corresponding editing site in the wild-type minigene, to mimic 100% editing at each site (Fig. 3b). Approximately 50% of *CCDC15* minigene-derived transcripts had exon 9 included, and 100% editing at site 2 significantly decreased exon 9 inclusion (Fig. 3b). Although mutation at site 1 weakly upregulated the inclusion level, concurrent mutations at sites 1 and 2 could still repress *CCDC15*-ex9 inclusion to a similar extent to mutation at site 2 alone (Fig. 3b), indicating that editing at site 2 causes a dominant repression on exon 9 inclusion. Intriguingly, removal of 13-bp core sequence at the edited region caused slightly more exon 9 included (Fig. 3b), suggesting that a weak ISS, which is a short RNA sequence promoting exon skipping by providing binding site(s) for splicing repressor(s), locates at the edited region. Editing at site 2 may enhance this ISS by affecting binding of splicing repressors.

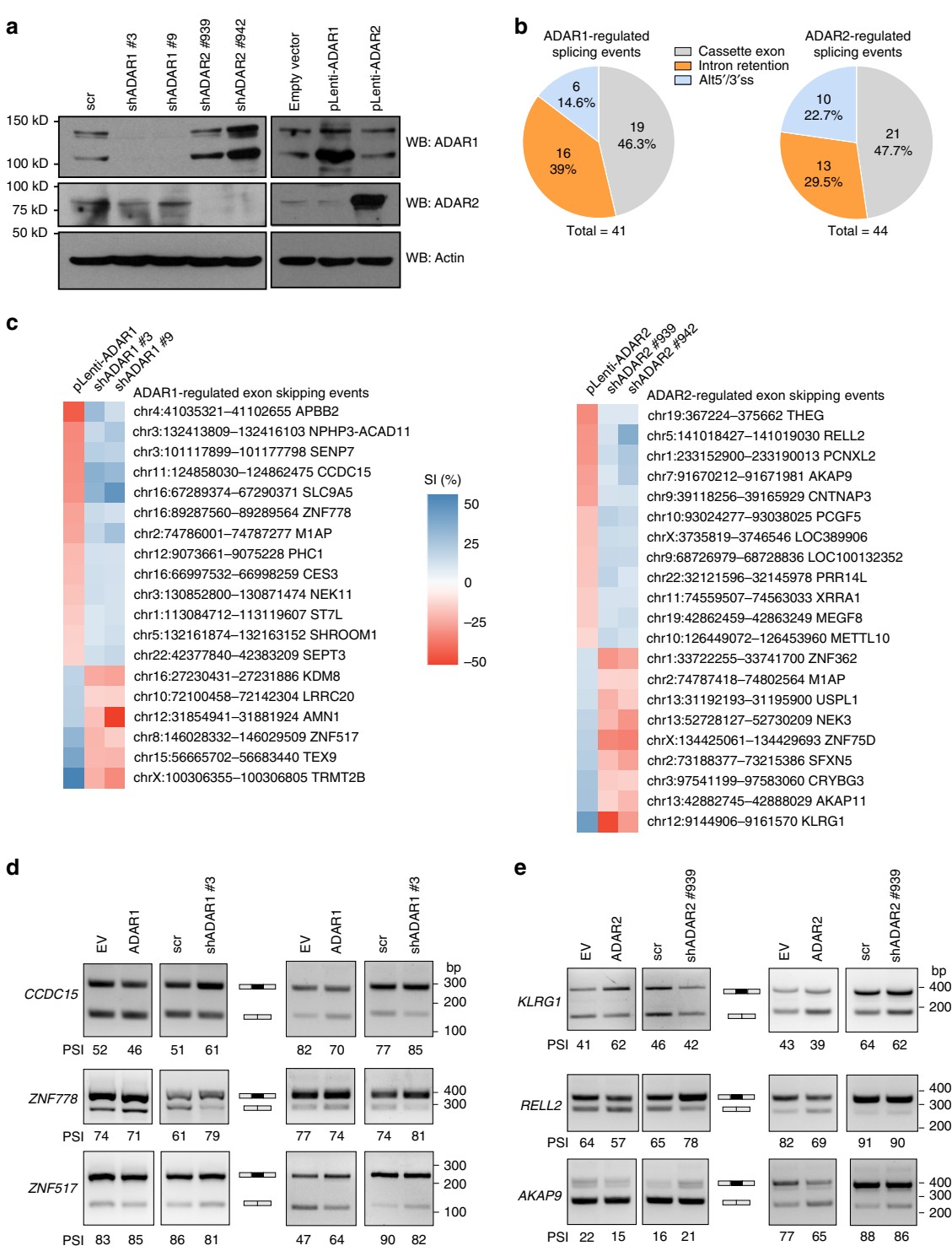

**Fig. 1 ADAR proteins regulate a subset of alternative splicing events. a** WB analyses of ADAR1 and ADAR2 proteins in EC109 cells that were stably knocked down (sh*ADAR1* #3 and #9; sh*ADAR2* #939 and #942; and scramble shRNA (scr)) or overexpressed (pLenti-*ADAR1*; pLenti-*ADAR2*; and empty vector control) for ADAR1 or 2, using lentiviral system. β-actin (actin) was used as a loading control. **b** Pie charts representing the number (and percentage) of each type of alternative splicing events affected by ADAR1 (left) and ADAR2 (right). **c** Heat maps showing the differentially spliced cassette exon events, upon knockdown and overexpression of ADAR1 (left) and ADAR2 (right). Splicing index (SI) is calculated by the ratio of inclusion junction reads to the sum of inclusion and skipping junction reads, and ΔSI indicates the difference in SI between ADARs knockdown/overexpression and their corresponding control samples. **d**, **e** RT-PCR analyses of representative ADAR1- **d** or ADAR2- **e** affected cassette exons in original RNA-Seq EC109 cells, as well as HEK293T cells. PSI, percent spliced in. Source data are provided as a Source Data file.

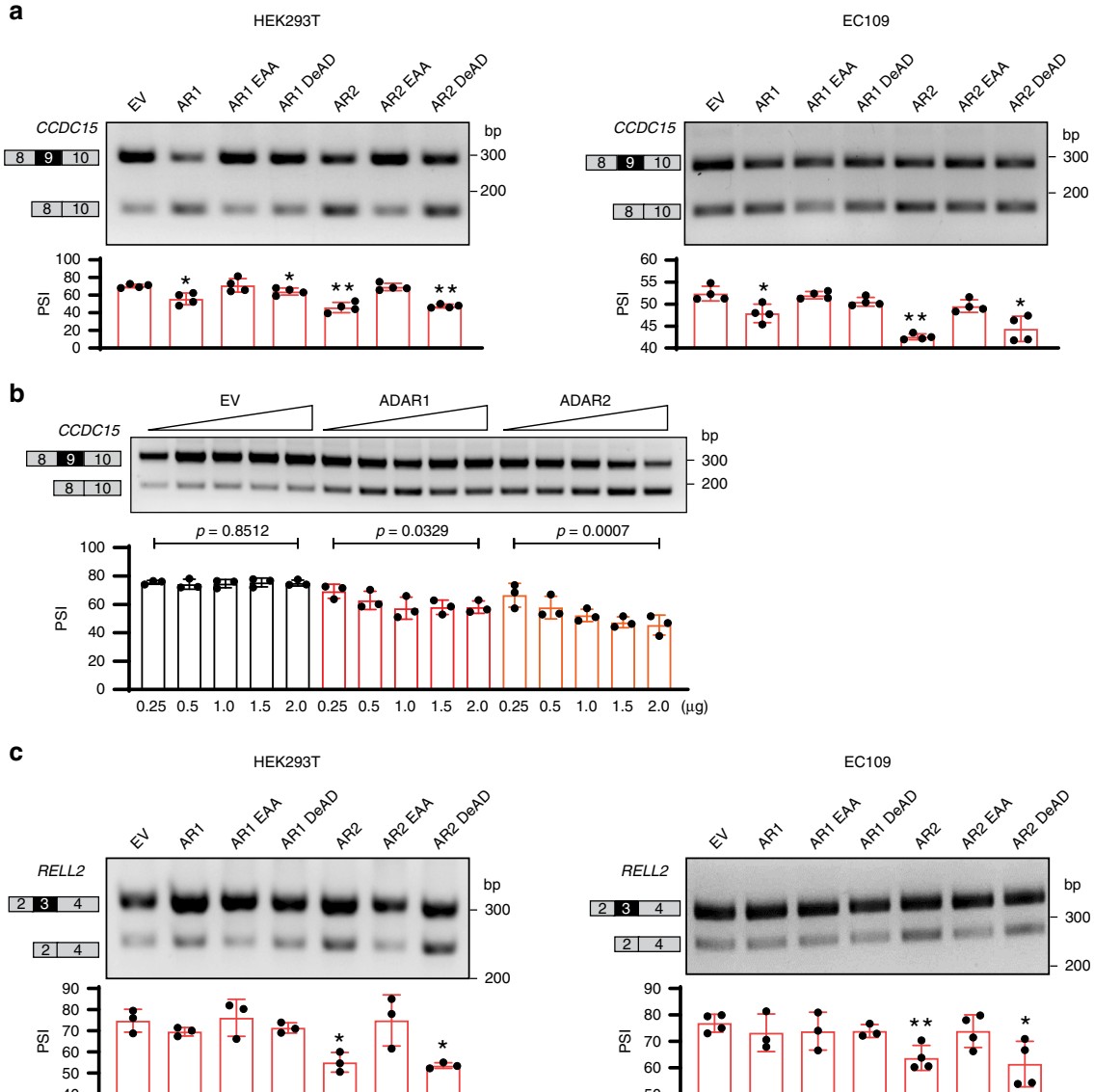

**Fig. 2 ADARs repress exon inclusion editing dependently and independently. a** RT-PCR analysis of *CCDC15*-ex9 inclusion in HEK293T (left) and EC109 (right) cells that were transfected with the indicated wild-type ADAR1 (AR1) or ADAR2 (AR2) and different mutant (EAA and DeAD) forms of expression constructs (*n* = 4 biological replicates for each). EV, empty vector. **b** RT-PCR analysis of *CCDC15*-ex9 inclusion in HEK293T that were transfected with the indicated amount of EV, *ADAR1*, or *ADAR2* construct (*n* = 3 biological replicates for each). Statistical significance of dose treatment is determined by linear regression. **c** RT-PCR analysis of *RELL2*-ex3 inclusion in the same samples as described in **a** (*n* = 3 or 4 biological replicates for each). **a**, **c** Data are presented as the mean ± S.D. of percent spliced in (PSI) values from biological replicates. Each dot represents a biological replicate. Statistical significance is determined by paired *t*-test (*\*P* < 0.05; *\*\*P* < 0.01). Source data are provided as a Source Data file.

In silico predictions using RBPmap[42] and Human Splicing Finder[43] suggest higher binding affinity of SRSF7 (serine/arginine-rich splicing factor 7) toward the edited sequence than wild-type sequence (Fig. 3c). To determine the effect of SRSF7 on *CCDC15*-ex9 inclusion in the presence of editing, HEK293T cells were co-transfected with wild-type or site 2-mutated minigene together with *SRSF7* expression construct. Upon SRSF7 over-expression, the repressive effect on *CCDC15*-ex9 inclusion was much stronger in the context of site 2 mutated than the wild-type minigene (Fig. 3d), implying that the regulatory effect of SRSF7 was augmented by editing. Consistently, RNA pulldown assay showed a stronger binding of SRSF7 to the edited intron 8 RNA probe than the wild-type probe; while hnRNPK, which was not predicted to differentially bind to edited *CCDC15* and included as a negative control, showed a similar binding affinity to both wild-type and edited probes (Fig. 3e). All these data suggest that

ADAR1 specifically edits a GA-rich ISS at intron 8 of *CCDC15*, which favors the binding of SRSF7 to the edited region and augments repression of exon 9 inclusion.

**Inter-intronic dsRNA is essential for splicing regulation.** As aforementioned, 100% editing at site 2 was insufficient to fully repress *CCDC15*-ex9 inclusion and DeAD mutant could not completely abolish ADAR1-mediated exon skipping (Figs. 3b and 2a), implying non-editing function of ADAR1 is involved in the regulation. In addition, editing-independent splicing regulation of ADAR2 on *CCDC15*-ex9 was further supported by the fact that point mutation at site 3 had no impact on exon 9 inclusion (Fig. 3b).

To unravel the precise editing-independent mechanisms responsible for ADAR1/2-mediated splicing regulation on *CCDC15*, we investigated ADAR1/2 acting region on *CCDC15*

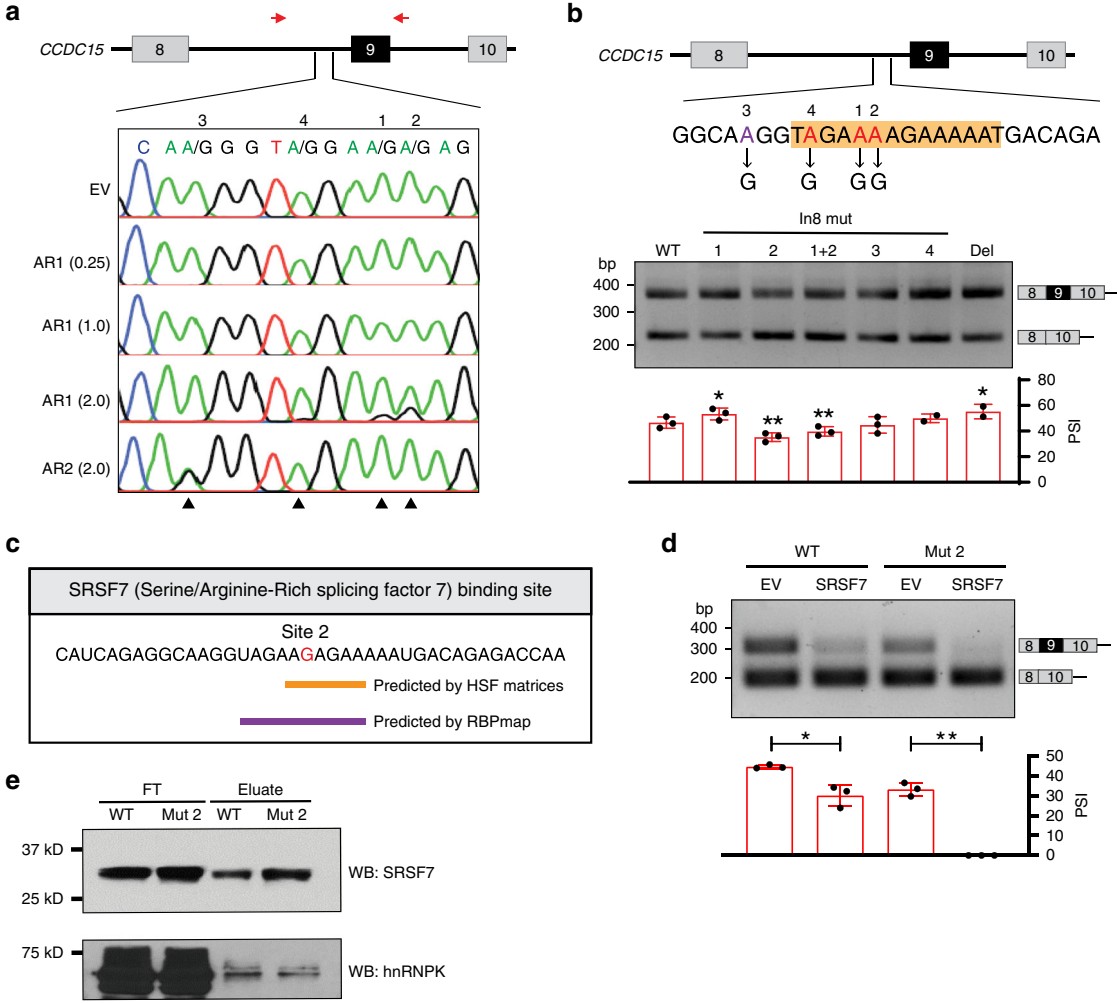

**Fig. 3 ADAR1-mediated editing of an ISS enhances SRSF7 binding for exon skipping. a** Sequence chromatograms illustrate the editing level of the indicated sites (1–4) at intron 8 of *CCDC15* pre-mRNA in HEK293T cells that were transfected with empty vector control (EV), *ADAR1* (0.25, 1.0, or 2.0 μg), or *ADAR2* (2.0 μg) expression construct. Black arrowhead indicates editing position. Red arrows show the location of primers used for PCR amplification. **b** Upper panel: schematic diagram of wild-type (WT) *CCDC15* exon 8–9–10 minigene. The positions where an A-to-G mutation was introduced are highlighted in red (sites 1, 2, and 4) and purple (site 3). The 13-bp region deleted in the Del minigene is shaded in orange. Lower panel: RT-PCR analysis of exon 9 inclusion of exogenous *CCDC15* transcripts in HEK293T cells that were transfected with the indicated WT or mutant minigenes (*n* = 3 or 2 biological replicates for each). 1 + 2 denotes both sites 1 and 2 were mutated from A to G. **c** In silico prediction of SRSF7 binding sites on the edited *CCDC15* pre-mRNA by Human Splicing Finder (orange line) and RBPmap (blue line). The edited nucleotide at site 2 is highlighted in red. **d** RT-PCR analysis of exon 9 inclusion of exogenous *CCDC15* transcripts in HEK293T cells that were co-transfected with WT or site 2-mutated (Mut 2) minigene together with EV or *SRSF7* expression construct (*n* = 3 biological replicates for each). **e** WB analysis of RNA pulldown products (eluate) shows binding of SRSF7 and hnRNPK protein to the WT or Mut 2 RNA probes. Sequence of the Mut 2 probe is shown in **c** and the WT probe is the same except site 2 that remains as an unedited adenosine. FT, flow-through. **b**, **d** Data are presented as the mean ± S.D. of percent spliced in (PSI) values from biological replicates. Each dot represents a biological replicate. Statistical significance is determined by paired *t*-test (*$P < 0.05$; **$P < 0.01$). Source data are provided as a Source Data file.

pre-mRNA by locating the editing site complementary sequence (ECS), which is essential for the formation of dsRNA structure for ADARs to bind and edit. Intron 9 of *CCDC15* was divided into three 300-nt long fragments (regions 1–3) for serial deletions in the wild-type *CCDC15* minigene (Fig. 4a). Upon co-transfection of each minigene and *ADAR1/2* plasmid, deletion of region 2 (Del 2) completely abolished repressive effects of ADAR1 and 2 on *CCDC15*-ex9 inclusion, indicative of the location of ADARs acting site in region 2 (Fig. 4b; left, Supplementary Fig. 3a, b). Further, we found that a short sequence (regions 4 and 5) at the 3′-end of region 2 is essential and sufficient for ADAR1-mediated editing and splicing regulation, as removal of either region eliminated ADAR1 effect (Fig. 4b; right, Fig. 4c, Supplementary Fig. 3c), signifying the presence of ECS.

Unlike ADAR1, although deletion of either region 4 or 5 abolished ADAR2-mediated editing (site 3), ADAR2 only lost its splicing-repressive effect in the absence of region 4 (Fig. 4b, c), suggesting that ADAR1 and 2 bind to the same dsRNA structure, but their binding regions on *CCDC15* are not 100% identical. As predicted by RNAfold[44], a nearly perfect dsRNA structure is formed between edited region at intron 8 and regions 4 and 5 at intron 9, with all three ADAR1-regulated sites in stem region (Fig. 4d; left). Removal of region 4 completely disrupted this dsRNA (Fig. 4d; middle), while deletion of region 5 retained part of dsRNA involving region 4 (Fig. 4d; right), allowing binding of ADAR2 as supported by a de novo editing site (site 5; Supplementary Fig. 3d).

Subsequently, to confirm the physical interaction between ADAR1/2 and *CCDC15*, RNA electrophoretic mobility shift assay

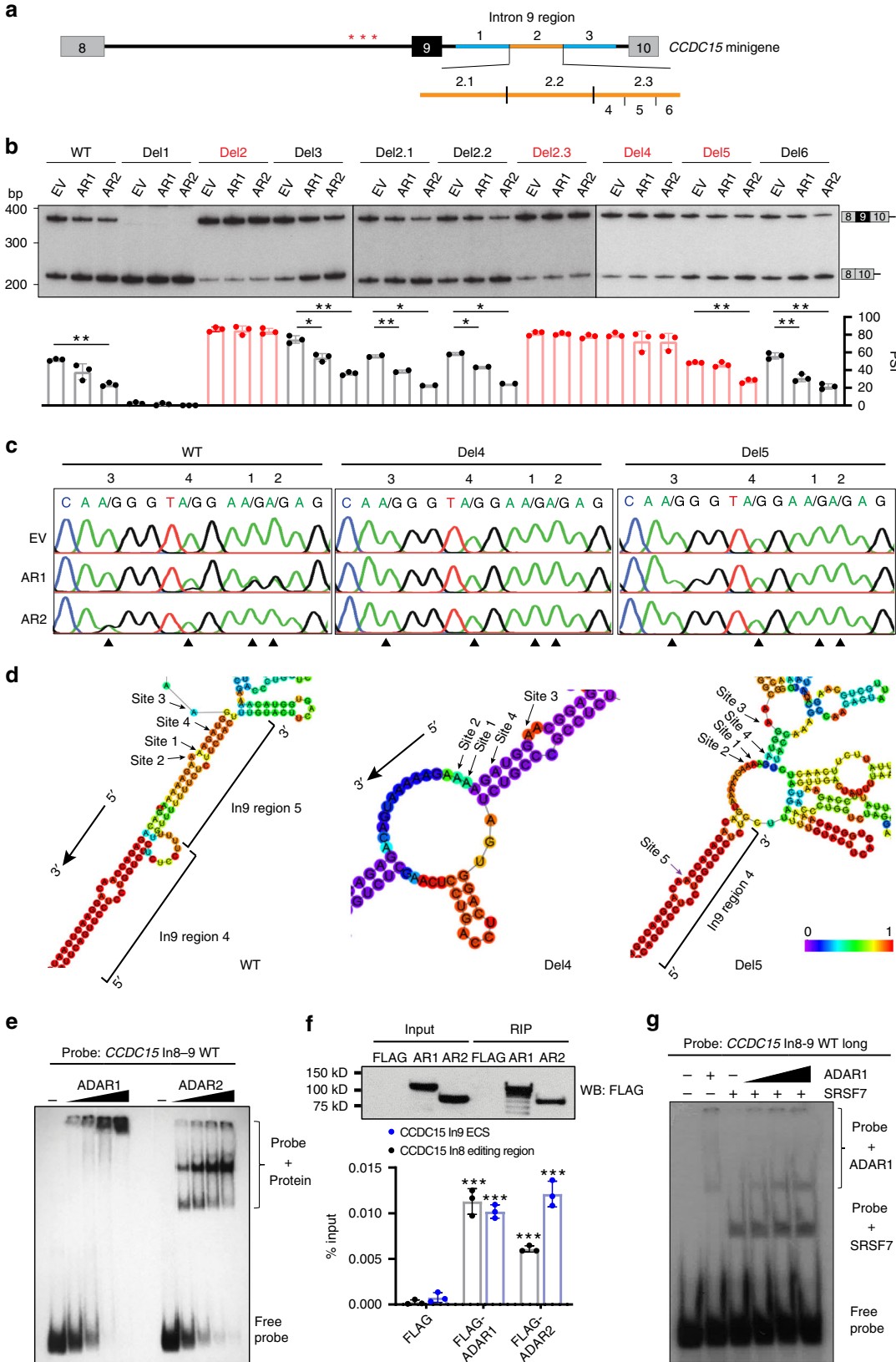

(REMSA) was carried out using an RNA probe that simulates *CCDC15* intronic dsRNA (Supplementary Fig. 3e; left). In the presence of recombinant ADAR1/2 protein, a dose-dependent band shift of protein–dsRNA complex was detected (Fig. 4e), and this band shift was attenuated by a mutant probe that forms an imperfect dsRNA (Supplementary Fig. 3e, f). Further, RNA immunoprecipitation (RIP) assays confirmed the association of ADAR1/2 with exogenous *CCDC15* dsRNA in vivo (Fig. 4f).

As ADAR1-mediated *CCDC15* editing enhances the binding of SRSF7 to the edited region, we next studied whether ADAR1

**Fig. 4 Inter-intronic dsRNA is essential for ADAR1/2 binding and splicing regulation. a** Schematic diagram illustrates serial deletions introduced into intron 9. Red asterisk represents editing site. **b** RT-PCR analysis of exon 9 inclusion of exogenous *CCDC15* transcripts in HEK293T cells that were co-transfected with the indicated minigene and overexpression construct (*n* = 3 or 2 biological replicates for each). Data are presented as the mean ± S.D. of percent spliced in (PSI) values from biological replicates. Each dot represents a biological replicate. Statistical significance is determined by paired *t*-test (*$P < 0.05$; **$P < 0.01$). **c** Sequence chromatograms illustrate the editing level of the indicated sites (1–4) in HEK293T cells that were co-transfected with the indicated minigene and overexpression construct. Black arrowhead indicates editing position. **d** In silico prediction of RNA secondary structure by RNAfold. Minimum free energy structures drawing encoding base-pair probabilities are shown. Base-pair probabilities are shown by a color spectrum. Arrow indicates the editing site. **e** REMSA analysis of binding of ADAR1 or ADAR2 protein to *CCDC15* transcripts in vitro, using a $^{32}$P-labeled RNA probe which simulates the dsRNA formed between introns 8 and 9 (*CCDC15* In8-9 WT) together with the increasing amount of recombinant ADAR1/2 protein. **f** RIP-quantitative PCR (qPCR) analysis of the binding of ADAR1 or ADAR2 protein to exogenous *CCDC15* transcripts (edited region in intron 8 and ECS in intron 9) in vivo (bottom panel). HEK293T cells were transfected with FLAG empty vector, FLAG-ADAR1, or FLAG-ADAR2, together with the wild-type *CCDC15* minigene, followed by RIP assay at 48 h post transfection. WB analysis of FLAG-RIP immunoprecipitates is shown in the top panel. Input indicates 1% of the total cell lysate. Data is presented as mean ± S.D. of %input derived from qPCR technical triplicates from a representative experiment. Each dot represents a technical replicate. Statistical significance is determined by unpaired, two-tailed Student's *t*-test (***$P < 0.001$). **g** REMSA analysis of the binding of SRSF7 to *CCDC15* transcripts in the absence or presence of ADAR1 protein in vitro, using a $^{32}$P-labeled *CCDC15* In8-9 WT long dsRNA probe. Source data are provided as a Source Data file.

binding to the *CCDC15* intronic dsRNA affects the binding of SRSF7 using a slightly longer *CCDC15* dsRNA probe (Fig. 4g, Supplementary Fig. 3g). An increased amount of ADAR1 protein did not cause a dose-dependent increase in the binding of SRSF7 to the probe (Fig. 4g), implying that ADAR1 binding is unlikely to affect the binding of SRSF7 to *CCDC15* dsRNA.

In sum, in addition to editing-dependent regulation of ADAR1, both ADAR1 and 2 proteins bind to the intronic dsRNA which forms between intron 8 editing region and a 46-nt sequence at intron 9 (spanning regions 4 and 5) of *CCDC15* pre-mRNA, which may preclude the access of splicing machineries to *CCDC15*-ex9, independent of RNA editing.

**ADAR2 binding to *RELL2* dsRNA prevents U2AF65 recruitment.** To study ADAR2 acting region on *RELL2* and decipher the underlying editing-independent mechanism of splicing regulation, we first searched for editing site(s) (as a footprint of ADAR2 binding) in exon 3 and surrounding regions. A weakly edited adenosine was observed at exon 3 in ADAR2-overexpressing HEK293T cells (Supplementary Fig. 4a).

Using the same strategy, a minigene containing *RELL2* exons 2, 3, and 4, and introns 2 and 3 was constructed and serial deletions were introduced into exon 3 and intron 3 (Fig. 5a). *RELL2* wild-type minigene-derived transcripts underwent the same splicing change and editing mediated by ADAR2 as endogenous *RELL2* (Supplementary Fig. 4b, c, Fig. 2c). Unexpectedly, none of the deletions had an impact on ADAR2-mediated splicing repression (Fig. 5b, Supplementary Fig. 4d, e), suggesting two possibilities: ADAR2 binds to somewhere outside this cassette or ADAR2 occupies multiple regions. To this end, we performed secondary structure prediction and found that the editing site is well situated at an almost perfect base-paired region formed by a GA-rich sequence (GAAGGAGA) at exon 3 and the polypyrimidine (Py)-tract at intron 2 (Fig. 5c; upper). Of note, there is another identical GA-rich sequence 14 nt downstream of the first GA-rich sequence, which could form dsRNA with the Py-tract when the first one is absent (Fig. 5c; middle, Fig. 5d). Therefore, G-to-C point mutations were introduced into both GA-rich sequences (double mutations) in the *RELL2* minigene to destroy the dsRNA structure (Fig. 5c; bottom). Consistently, double mutations completely abolished the repressive effect of ADAR2 on *RELL2*-ex3 inclusion (Fig. 5d, Supplementary Fig. 4e). *RELL2* dsRNA probe consisting of BPS, Py-tract, and full-length (FL) exon 3 was subjected to REMSA analysis (Fig. 5e, Supplementary Fig. 4f; left panel). A gradual increase in band shift of ADAR2–*RELL2* Py-ex3 complex was observed upon addition of ADAR2; while this band shift was attenuated by the mutations of double GA-rich

sequences, verifying direct interaction of ADAR2 to this *RELL2* dsRNA in vitro (Fig. 5e, Supplementary Fig. 4f, g). RIP assays further confirmed the direct association of ADAR2 to exogenous *RELL2* Py-ex3 dsRNA in vivo (Fig. 5f).

The Py-tract is essential for U2AF65 binding to define downstream exon[45,46], presumably that binding of ADAR2 to the Py-tract at intron 2 may prevent U2AF65 from binding, leading to exon skipping. To this end, we conducted in vitro RNA–protein binding assay by UV crosslinking and found that in the presence of ADAR2, binding of U2AF65 to the probe was much weaker when compared to U2AF65 alone (Fig. 5g). In addition, there was no obvious effect on the binding of ADAR2 to the Py-ex3 dsRNA in the presence of U2AF65 (Supplementary Fig. 4h). Consistently, RNA pulldown assay showed that ADAR2 blocked binding of U2AF65 to the Py-ex3 probe in a dose-dependent manner (Fig. 5h). Altogether, these data demonstrated that binding of ADAR2 to dsRNA formed between GA-rich sequences at exon 3 and the Py-tract at intron 2 blocks the access of U2AF65 for exon recognition, rendering skipping of *RELL2* exon 3.

**GA-rich motif may be involved in ADARs-regulated splicing.** As GA-rich sequences are involved in the regulation of both *CCDC15*-ex9 and *RELL2*-ex3 splicing, we went on to investigate whether binding of ADARs to GA-rich sequences could be involved in other ADARs-regulated splicing events. Using Multiple Em for Motif Elicitation (MEME)[47], we conducted an unbiased analysis of binding motifs that are enriched in pre-mRNA sequences of ADARs-regulated cassette exons and their proximal flanking regions (Supplementary Data 1). To identify motif(s) specifically involved in ADARs-mediated splicing regulation, sequences of non-ADARs-regulated cassettes exons detected by our RNA-Seq were also included for the analysis (Supplementary Data 1). Of four motifs that were found to be enriched in ADARs-regulated cassette exons and flanking sequences, a GA-rich motif was not detected in non-ADARs-regulated counterparts (Fig. 6a, b). Importantly, ~66% of ADAR1/2-regulated events (37/56) contain at least one GA-rich motif within exon or flanking sequences (Fig. 6a). These findings support and expand our earlier observation that GA-rich sequences were involved in the formation of dsRNA and binding of ADAR1/2 to *CCDC15* and *RELL2* transcripts (Figs. 4d and 5c). Further, RNA secondary structure predictions for several candidates indicated that GA-rich motif is well situated in dsRNAs (Fig. 6c). Detection of editing sites located in close proximity to the GA-rich motif in *AKAP11* and *KLRG1* transcripts confirmed the binding of ADARs to dsRNA involving GA-rich sequence

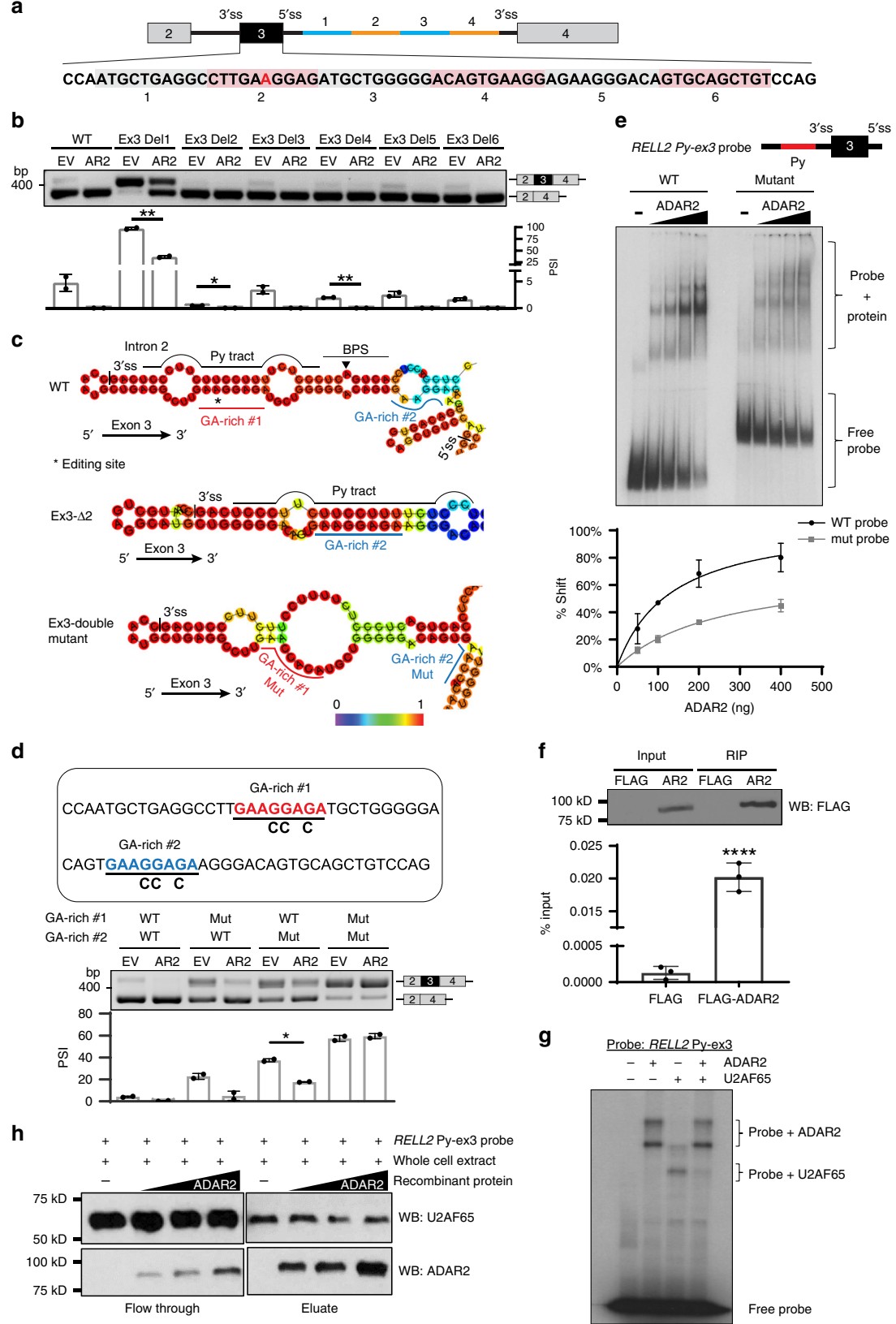

(Fig. 6d). Intriguingly, for *M1AP* and *KLRG1*, the GA-rich motif and Py-tract were predicted to form base-pairs in dsRNAs (Fig. 6c), suggesting that binding of ADARs to dsRNA formed between the GA-rich motif and Py-tract may be a common splicing regulatory mechanism. Altogether, a GA-rich motif may

be generally involved in dsRNA formation for ADARs binding and splicing regulation.

**Skipping of *RELL2* exon 3 leads to nonsense-mediated decay.** We next investigated the potential biological consequences of

**Fig. 5 ADAR2 binding to dsRNA that involves the Py-tract blocks U2AF65. a** Schematic diagram of *RELL2* minigene and serial deletions in exon 3 and intron 3. Editing site is highlighted in red. **b** RT-PCR analysis of exon 3 inclusion of exogenous *RELL2* transcripts in HEK293T cells that were co-transfected with the indicated minigene and overexpression construct ($n = 2$ biological replicates for each). **c** In silico prediction of RNA secondary structure by RNAfold. Base-pair probabilities are shown by a color spectrum. Asterisk indicates the editing site. **d** RT-PCR analysis of exon 3 inclusion of exogenous *RELL2* transcripts in HEK293T cells that were co-transfected with the indicated minigene and overexpression construct ($n = 2$ biological replicates for each). GA-rich sequences #1 and #2 are highlighted in red and blue. **e** REMSA analysis of the binding of ADAR2 protein to *RELL2* transcripts in vitro, using $^{32}P$-labeled wild-type or mutant *RELL2* Py-ex3 dsRNA probe with the increasing amount of recombinant ADAR2 protein. %Shift is calculated as shift band intensity over the sum of free probe and shift band intensities. **f** RIP-qPCR analysis of the binding of ADAR2 protein to exogenous *RELL2* transcripts in vivo. WB analysis of FLAG-RIP immunoprecipitates is shown in the top panel. Input indicates 1% of the total cell lysate. Data are presented as mean ± S.D. of % input derived from qPCR technical triplicates from a representative experiment. Each dot represents a technical replicate. Statistical significance is determined by unpaired, two-tailed Student's *t*-test (****$P < 0.0001$). **g** In vitro RNA–protein binding assay by UV crosslinking demonstrated the binding of U2AF65 to the *RELL2* dsRNA probe, in the absence or presence of ADAR2 protein. **h** RNA pulldown assay detected the binding of U2AF65 protein to the dsRNA probe, in response to the addition of increasing amount of ADAR2 protein. WB analysis of U2AF65 and ADAR2 proteins in RNA pulldown (eluate) products and flow-through fractions. **b, d** Data are presented as the mean ± S.D. of percent spliced in (PSI) values from biological replicates. Each dot represents a biological replicate. Statistical significance is determined by paired *t*-test (*$P < 0.05$; **$P < 0.01$). Source data are provided as a Source Data file.

ADARs-mediated splicing regulation. Alternative splicing of a cassette exon generally preserves reading frame and produces different protein isoforms, such as *CCDC15*-ex9. Nonetheless, inclusion of a poison exon or skipping of an essential exon, such as *RELL2*-ex3, could disrupt reading frame and result in nonsense-mediated decay (NMD), known as AS-NMD that can regulate gene expression.

Skipping of *RELL2* exon 3 results in a frameshift and creates a pre-mature termination codon at the beginning of exon 5, which locates 368 nt upstream of the last exon–exon junction, making the skipped transcripts susceptible to NMD degradation (Fig. 7a). Inhibition of NMD by cycloheximide (CHX) treatment or silencing of *UPF1*, a core NMD factor[48,49], caused a remarkable increase in the level of exon 3-skipped transcript (Fig. 7b), suggesting that exon 3 skipping produces an NMD-sensitive isoform. Consistently, in response to CHX treatment and *UPF1* knockdown, NMD degradation of exon 3-skipped isoform was inhibited, resulting in an increase in total *RELL2* transcript level (Fig. 7c). In addition, expression level of total *RELL2* mRNA was gradually decreased upon dose-dependent overexpression of ADAR2 (Fig. 7d), implying that ADAR2 represses exon 3 inclusion and triggers NMD-mediated degradation of transcripts.

**ADARs-regulated splicing is implicated in tumorigenesis.** To understand the functional importance of ADARs-mediated splicing changes in cancer, we examined the splicing pattern of *CCDC15*-ex9 and *RELL2*-ex3 in 33 matched pairs of primary ESCC and non-tumor (NT) samples. Based on the RNA expression levels of ADAR1 and ADAR2 as detected by quantitative polymerase chain reaction (qPCR), ESCC tumors were divided into four groups: ADAR(1/2)^high and ADAR(1 and 2)^normal/low or ADAR2^high, and ADAR2^normal/low (Supplementary Fig. 5a). We observed that 26% (6/23) of ADAR(1/2)^high ESCC tumors showed decreased level of *CCDC15*-ex9 inclusion than their matched NT samples; while this change was observed in in only 10% (1/10) of ADAR(1 and 2)^normal/low tumors (Fig. 8a, Supplementary Fig. 5b). Moreover, *CCDC15*-ex9 inclusion were upregulated in 60% (20/33) of ESCC tumor samples, implying the potential oncogenic role of *CCDC15*-ex9 included isoform. As for *RELL2*, ~50% (5/11) of ADAR2^high tumors and 27% (5/22) of ADAR2^normal/low demonstrated downregulation of exon 3 inclusion (Fig. 8b, Supplementary Fig. 5c). These data suggest that aberrant splicing of both *CCDC15* and *RELL2*, which is at least partially due to differentially expressed ADARs in tumors, might be implicated in ESCC tumorigenesis.

Next, EC109 cells with lentivirus-mediated stable knockdown of exon 9-included *CCDC15* showed lower frequency of foci formation and formed less colonies in soft agar (Fig. 8c, d, Supplementary Fig. 5d). To ensure the observed functional changes were not due to off-target effects of one single short hairpin RNA (shRNA), EC109 cells transduced with two additional shRNAs against total *CCDC15* also demonstrated less aggressive phenotypes than control cells (Supplementary Fig. 5e, f). Xenograft tumors derived from knockdown cells grew much slower than those from control cells (Fig. 8e). Consistently, overexpression of exon 9-included CCDC15 (FL) promoted tumorigenesis in EC109 cells, as detected by in vitro and in vivo tumorigenicity assays (Fig. 8f–h, Supplementary Fig. 5g).

For *RELL2*, as its expression was very low in EC109, possibly due to the fact that exon 3-skipped isoform is sensitive to NMD. Therefore, exon 3-included *RELL2* was stably overexpressed in EC109 cells using a lentiviral system (Supplementary Fig. 5h). Overexpression of RELL2 contributed to less tumorigenic phenotypes, as detected by both in vitro foci formation and soft agar assays, as well as in vivo xenograft assay (Fig. 8i–k). All these data suggest that exon 9-included *CCDC15* and exon 3-included *RELL2* isoforms are oncogenic and tumor suppressive, respectively. More importantly, ADARs-mediated splicing changes indeed affect tumorigenesis, not merely byproducts of ADARs editing and binding.

**Discussion**

As previously appreciated, modification of canonical *cis*-acting elements by RNA editing is unlikely to be a widespread mechanism of editing-regulated splicing changes[27]; instead, RNA editing enzymes ADARs and splicing machineries may contribute to their crosstalk through acting on the same dsRNA regions[20,50]. However, it remains poorly understood whether splicing modulations mediated by secondary structural changes of pre-mRNAs are arising from ADARs binding or editing. Despite both ADAR1 and ADAR2 are primary RNA editing enzymes, the majority of research studies focused on the role of ADAR1, but not ADAR2, in regulating splicing. In this study, we identified approximately a hundred high-confidence alternative splicing events by applying very stringent criteria. Of note, we found that ADAR1 and ADAR2 regulate cassette exons in both directions. However, some previously reported crosstalk splicing events, such as *STAT3*, *hnRNPR*, and *hnRNPLL* were not identified by our RNA-Seq analysis[20,23,24,27], presumably due to differential expressions of editing regulators in different types of cells and the resultant absence or low level of RNA editing at particular targets[34,51]. This

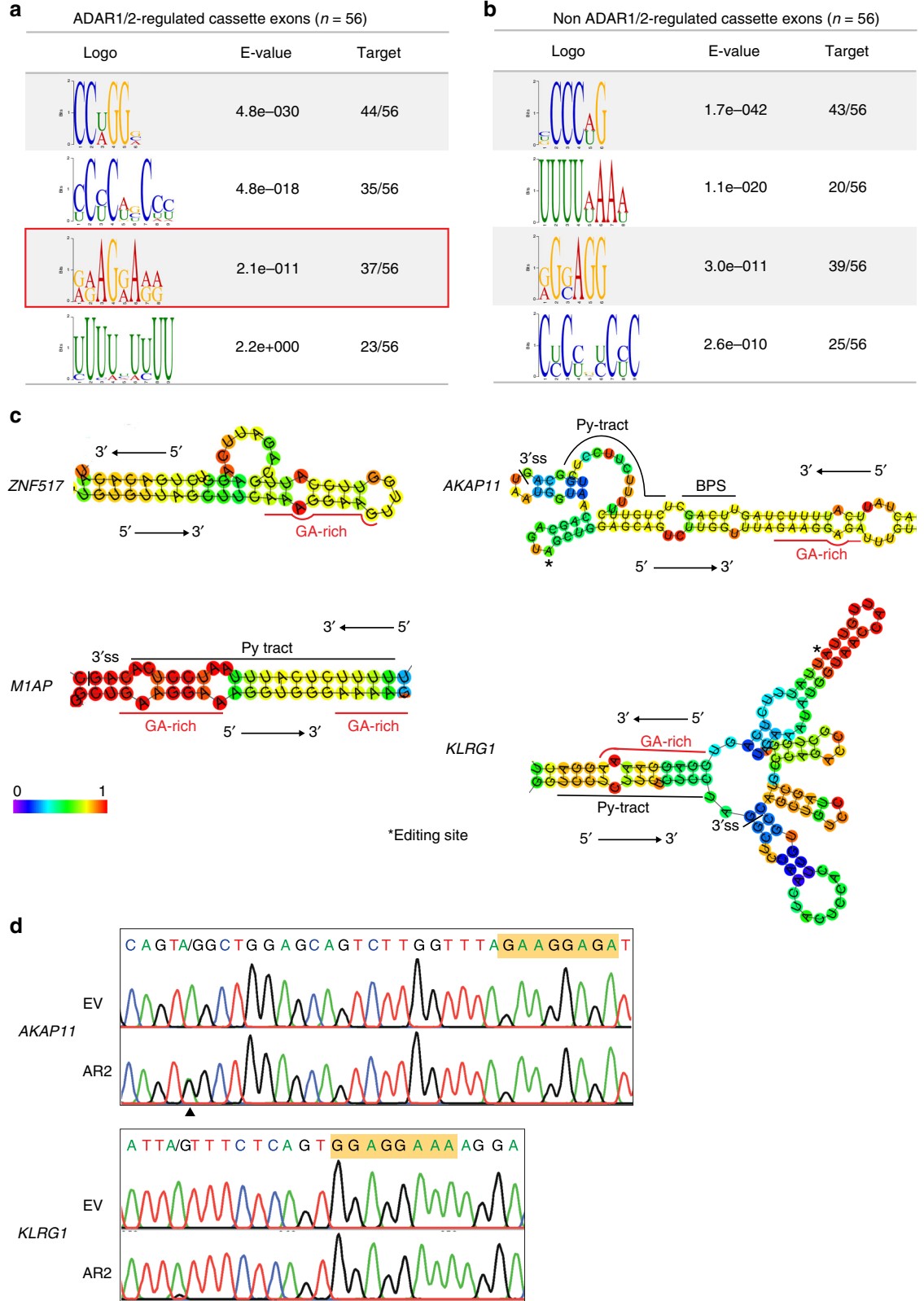

**Fig. 6 Involvement of GA-rich motif(s) in ADARs-mediated splicing regulation. a, b** The top four motifs enriched in ADARs-regulated cassette exons **a** and non-ADARs-regulated cassette exons **b** as provided by MEME. Height of letter depicts occurrence of nucleotides at specific position. *E*-value represents statistical significance of the motif. Target represents the number of cassette exons containing a particular motif over total input number. **c** RNA secondary structure predictions for pre-mRNA sequences of representative ADARs-regulated cassette exons by RNAfold. Base-pair probabilities are shown by a color spectrum. Red line indicates the GA-rich motif. Asterisk denotes editing site. **d** Sequence chromatograms illustrate editing site adjacent to GA-rich motif of *AKAP11* and *KLRG1* upon ADAR2 overexpression in HEK293T cells. GA-rich motif is shaded in orange. Black arrowhead indicates editing position.

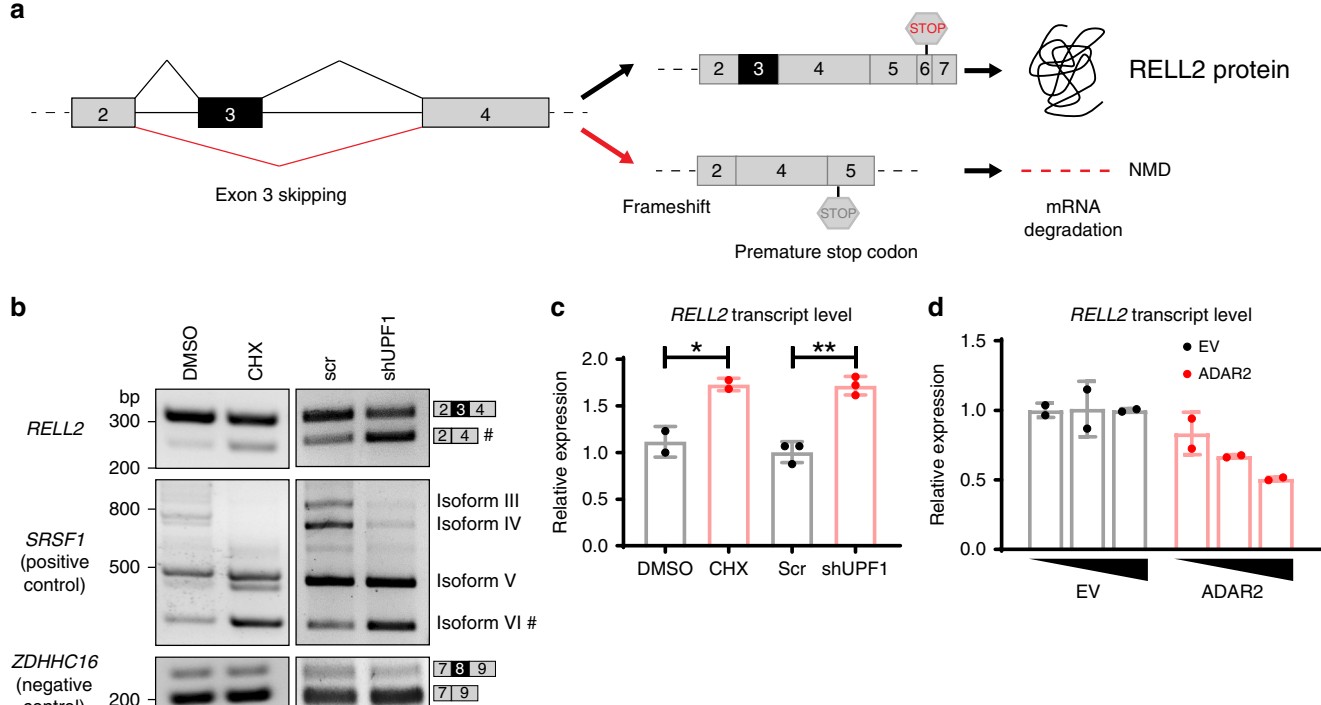

**Fig. 7 Skipping of *RELL2* exon 3 results in AS-NMD. a** Schematic diagram showing skipping of *RELL2* exon 3 may trigger NMD. With exon 3 included, *RELL2* pre-mRNA has a stop codon at the end of exon 6, which is then translated to a functional protein. In contrast, skipping of exon 3 results in a frameshift and creates a pre-mature termination codon at the beginning of exon 5, which locates 368 nt upstream of the last exon–exon junction. **b** RT-PCR analysis of different isoforms of the indicated transcripts in HEK293T cells upon inhibition of NMD. Cells were transfected with scramble shRNA (scr) or shRNA against *UPF1* (sh*UPF1*) for *UPF1* knockdown, or treated with DMSO or CHX. *SRSF1* or *ZDHHC16* serves as a positive or negative control, respectively, to ensure successful inhibition of NMD. Number sign indicates the NMD-sensitive isoform. **c** qPCR analysis of total *RELL2* transcript level in HEK293T cells with the indicated treatment. Data are presented as the mean ± S.D. of relative expression derived from qPCR technical triplicates or duplicates from a representative experiment. Each dot represents a technical replicate. Statistical significance is determined by unpaired, two-tailed Student's $t$-test (*$P <$ 0.05; **$P <$ 0.01). **d** qPCR analysis of total *RELL2* transcript level in HEK293T cells that were transfected with the increasing amount of EV (empty vector) or *ADAR2* construct. Data are presented as the mean ± S.D. of relative expression derived from qPCR technical duplicates from a representative experiment. Each dot represents a technical replicate. Source data are provided as a Source Data file.

could also imply that the crosstalk between RNA editing and splicing machineries may be dynamically regulated under certain conditions or diseases, such as cancer.

Their crosstalk via editing-mediated sequence alteration(s) largely depends on editing frequency and location of editing site. Although not common, editing at canonical *cis*-acting elements can result in dramatic changes in splicing and gene expression, since it can completely abolish or create exon recognition. On the other hand, effect of editing at auxiliary *cis*-acting elements (e.g., ISS and ESE) may vary depending on strength of original elements or sequence compositions that determine binding affinity toward splicing factors. Herein, we unraveled that ADAR1 binds to an intronic dsRNA of *CCDC15* formed between introns 8 and 9, and edits an ISS in intron 8, which could improve the strength of an existing ISS to repress exon 9 inclusion by enhancing binding of splicing repressor SRSF7 (Fig. 9). Moreover, it should be noted that for both *CCDC15* and *RELL2*, their edited sequences by ADAR1 or ADAR2 are GA-rich (e.g., GAAAAG or GAAGGA). Despite the low impact of editing on splicing due to the lack of substantial modification of canonical *cis*-acting elements by editing, ADARs-mediated editing of GA-rich sequences within auxiliary *cis*-regulatory elements may be an important mechanism of editing-mediated splicing changes. SR proteins, such as SRSF1, SRSF2, SRSF4, SRSF7, and SRSF10, which generally act as splicing activators, tend to interact with purine-rich sequences[52–56]. In addition, splicing repressor hnRNPA1 shows a binding preference toward

UAGG sequence[57]. Therefore, editing may modulate sequence binding affinity for these splicing factors. Another mechanism through which editing may change splicing is by affecting the RNA secondary structure stability and eventually altering the accessibility of splice sites, which is unlikely in the case of *CCDC15*.

Our study also provided extensive experimental evidence for previously proposed editing-independent crosstalk mechanisms, in which ADARs may alter splicing either by competing binding to dsRNA regions with splicing machineries or by recruiting splicing factors[58,59]. It was unanticipated that both editing and dsRNA-binding capabilities of ADARs are involved in the repression of *CCDC15*-ex9 inclusion. More importantly, we discovered that ADAR1 and ADAR2 could bind to the same dsRNA region of a target pre-mRNA and influence splicing in the same direction. However, whether they regulate splicing through same or distinct mechanisms, remains a topic for further investigation. Intriguingly, our study represented a mechanism by which binding of ADAR2 to *RELL2* dsRNA formed between GA-rich sequences at exon 3 and its upstream uridine-rich Py-tract sterically precludes access of U2AF65 to 3′ss for exon recognition (Fig. 9). This mechanism is very much similar to that employed by common splicing factors PTBP1 and hnRNPC, in which they bind to the Py-tract and block U2AF65 from binding[60,61]. Of note, we observed >60% of ADAR1/2-regulated cassette exons containing at least one GA-rich motif within exon or proximal flanking sequences. Similar to *RELL2*, some targets, such as

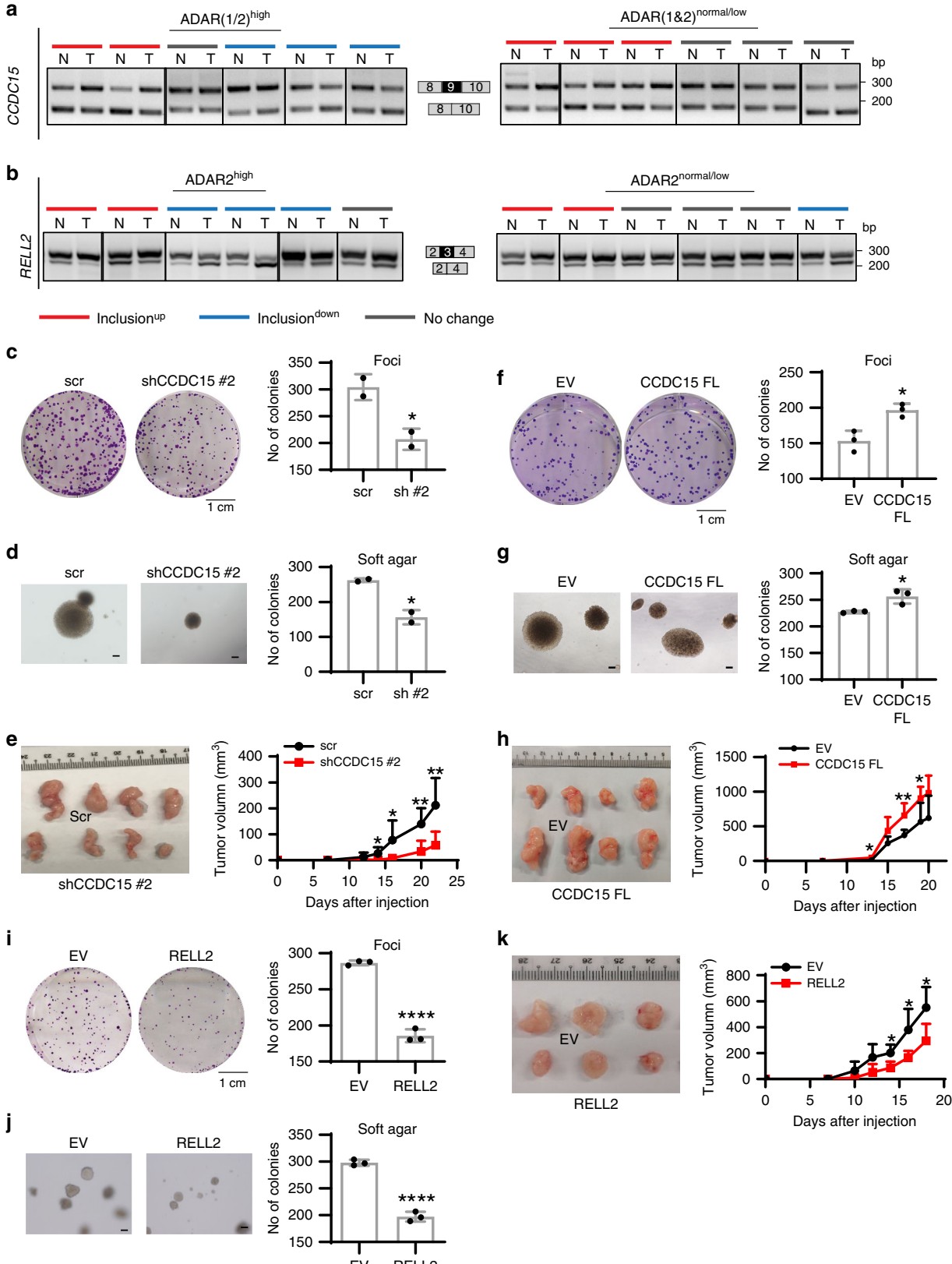

*M1AP* and *KLRG1* were predicted to form dsRNA between the GA-rich motif and Py-tract. These findings suggested a common editing-independent crosstalk mechanism, in which ADARs directly competes with SR or hnRNP proteins or spliceosome components, such as U2AF65 for binding to the dsRNA forming GA-rich *cis*-acting elements or complementary neighboring sequences, such as Py-tract and 3′ss, respectively. In addition, from our RNA-Seq data, ADAR proteins regulate splicing in both directions; however, mechanisms determining regulatory direction of ADARs remain unclear. We propose that similar to

**Fig. 8 ADARs-mediated splicing changes affect tumorigenesis. a, b** RT-PCR analysis of *CCDC15*-ex9 **a** or *RELL2*-ex3 **b** splicing pattern in 12 representative matched pairs of primary ESCC and NT samples. Patients demonstrating ≥1.5-fold higher expression of ADAR1 or/and ADAR2 in tumors than their matched NT tissues were classified as ADAR(1/2)$^{high}$ group; while the remaining cases were included into ADAR(1 and 2)$^{normal/low}$ group. Patients demonstrating ≥1.5-fold higher expression of ADAR2 in tumors than their matched NT tissues were classified as ADAR2$^{high}$ group; while the remaining cases were included into ADAR2$^{normal/low}$ group. Tumor samples with >5% increase in exon inclusion as compared to NT samples were defined as inclusion$^{up}$ group; while tumor samples with >5% decrease in exon inclusion as compared to NT samples were defined as inclusion$^{down}$ group. Red, blue, and gray lines indicate inclusion$^{up}$, inclusion$^{down}$, and no change cases, respectively. **c, f, i** Quantification of foci formation induced by the indicated cells ($n = 2$ or 3 biological replicates for each). Scale bar, 1 cm. **d, g, j** Quantification of colonies formed in soft agar induced by the indicated cells ($n = 2$ or 3 biological replicates for each). Scale bar, 200 μm. **e, h, k** Representative image of xenograft tumors derived from the indicated cells ($n = 7$ **e**, 6 **h**, or 5 **k** mice per group) at end point (left panel). Growth curves of tumors derived from the indicate cells are shown in right panel. **c, d, f, g, i, j** Data are presented as the mean ± S.D. of biological replicates from a representative experiment. Each dot represents a biological replicate. Statistical significance is determined by unpaired, two-tailed Student's *t*-test (*$P < 0.05$; ****$P < 0.0001$). **e, h, k** Data are presented as the mean ± S.D. of tumor volumes. Statistical significance is determined by unpaired, two-tailed Student's *t*-test (*$P < 0.05$; **$P < 0.01$). Source data are provided as a Source Data file.

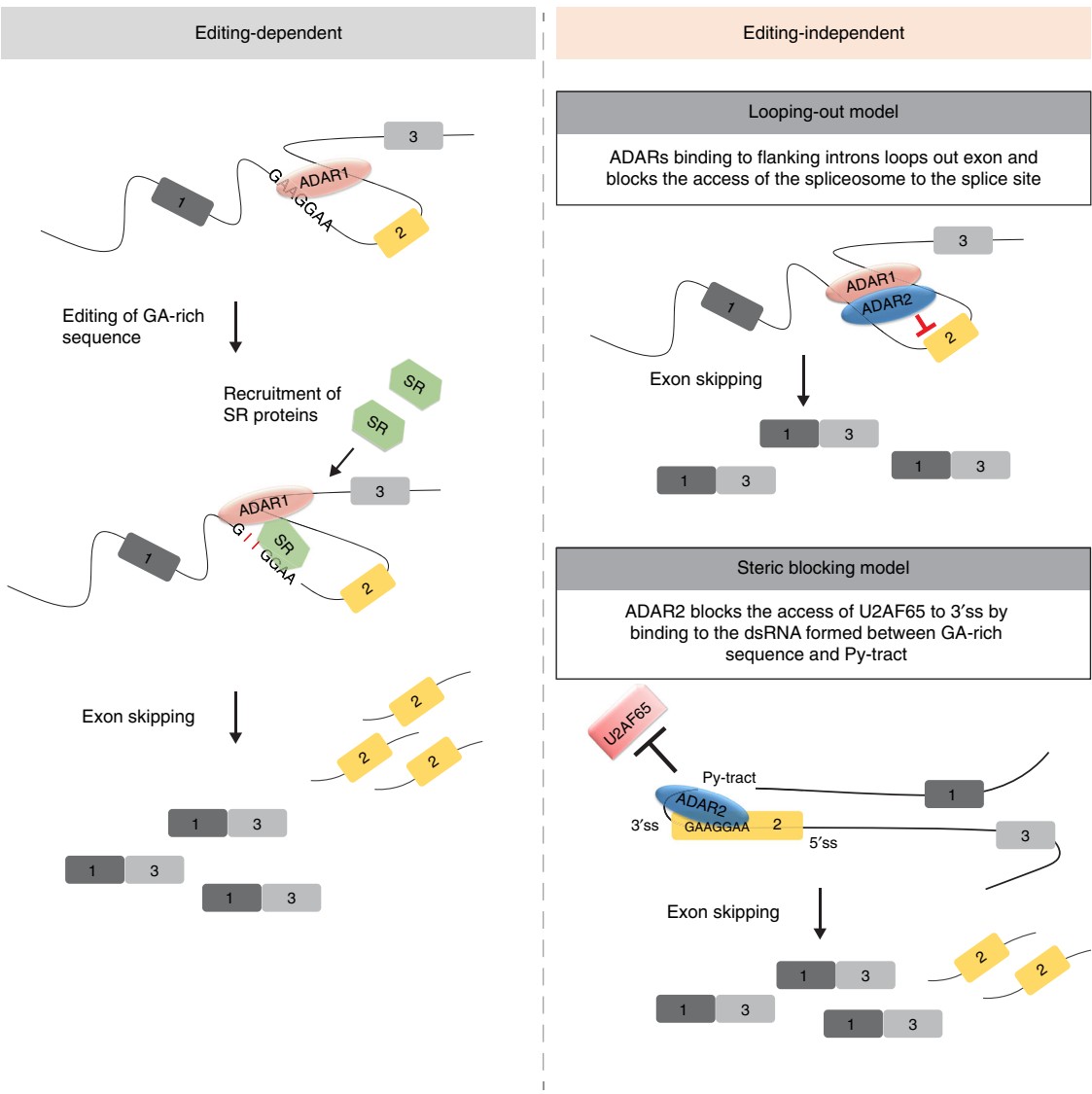

**Fig. 9 Schematic mechanistic diagram of ADARs-regulated alternative splicing.** Conversion of adenosine to inosine by ADAR1 at a GA-rich sequence proximal to cassette exon of a target transcript recruits SR proteins or other splicing factors to repress exon inclusion. Independent of their editing functions, ADAR1 and ADAR2 proteins can bind to inter-introns of a target transcript, thereby looping out the exon and blocking the access of the spliceosome to the splice site. Alternatively, ADAR2 blocks the access of U2AF65 to the 3′ splice site by binding to the dsRNA formed between the GA-rich sequence and Py-tract.

hnRNPs, the relative location of ADARs acting region to alternative exon could be one of the determinants[62]. As such, binding of ADAR to exon–intron junction blocks exon recognition by the spliceosome, while binding of ADAR to intra-intronic dsRNA may promote exon inclusion by enhancing intron definition.

RNA editing and splicing have been extensively studied, and dysregulation of both processes and their machineries impose considerable impact on cancer development[16,28–38]. However, there are very limited understanding of the role of their crosstalk in cancer. In this study, we found that these ADARs-regulated splicing changes per se are implicated in tumorigenesis. ESCC tumors expressing high level of ADAR1 or/and ADAR2 demonstrated more *CCDC15*-ex9 repression, indicative of regulatory roles of ADAR1 and ADAR2. Nonetheless, due to differentially expressed ADARs and other possible splicing factors in tumors, *CCDC15*-ex9 was preferentially included in over 50% of primary ESCC tumors and more importantly, exon 9-included CCDC15 was found to be oncogenic. Using PANDA[63], a protein function analysis tool, FL CCDC15 (exon 9 included) is predicted to have multiple functions, including nucleic acid binding, helicase, ATPase, and transcriptional repressor. In addition, according to the Human Protein Atlas, CCDC15 protein mainly expresses on cell membrane or in cytoplasm. Altogether, CCDC15 may be involved in signaling pathways that can promote tumorigenesis via regulating gene expression. Skipping of *CCDC15* exon 9 is likely to disrupt its functional domain and inhibit the putative gene regulation. On the other hand, ADAR2-mediated splicing repression of *RELL2*-ex3 could decrease *RELL2* expression via NMD and promotes tumorigenesis. Previous studies have demonstrated antimetastatic and proapoptotic functions of RELL2 in breast cancer cells and HEK293T cells, respectively[64,65], supporting a tumor suppressive role of RELL2. Although ADAR2 is well characterized as a tumor suppressor in several types of cancer[16,32,66], this regulation of *RELL2*-ex3 inclusion may open up an alternative functional role of ADAR2 in cancer. Nonetheless, we could not deny the possibility that the splicing outcomes of these crosstalk events may be determined by a combinatory effect of ADARs and other splicing regulators in the context of cancer. By knowing the involvement of a GA-rich motif in ADARs binding and ADARs-mediated splicing regulation, we are able to identify spliced transcripts specifically regulated by ADARs. The existence of these novel transcripts arising from interplay between editing enzymes and splicing machineries increases the complexity of transcriptome, and aberrantly regulated crosstalk events may contribute to human diseases such as cancer.

## Methods

**Cell lines**. EC109 cells (kindly provided by Professor Tsao (Head, Department of Anatomy, The University of Hong Kong)) and HEK293T cells (ATCC) were maintained in RPMI (Roswell Park Memorial Institute) 1640 and DMEM (Dulbecco's Modified Eagle Medium) media (Biowest) supplemented with 10% fetal bovine serum, respectively, at 37 °C, 5% $CO_2$.

**Analysis of RNA-Seq data**. EC109 cells were stably knocked down or overexpressed of ADAR1 or ADAR2 by lentiviral-mediated shRNA silencing or overexpression system. RNA extraction was performed using RNeasy Mini kit (Qiagen) and subjected to strand-specific paired-end sequencing (BGI). Read mapping was conducted using STAR[67] with the reference human genome version 19 as the annotation library. Read statistics are summarized in Supplementary Table 1.

Splicing analysis was performed using in-house bioinformatic pipeline[68,69]. Briefly, mapped reads were filtered and classified into the following three groups based on their distance and location around the annotated splice sites: (1) reads mapped to exon–exon junctions, (2) reads bridging exon–intron junctions, and (3) reads mapped completely to introns. The reads in group 3 were used for intron retention coverage and depth calculations. The reads in groups 1 and 2 were used for the calculation of SI for each splice site for the selection of potential differential splicing candidates. To identify the various types of splicing events, including exon skipping, intron retention, and alternative splice sites, the SI calculation was adjusted for each type of splicing events[68]. All potential splicing candidates were classified into either known/annotated alternative splicing or novel splicing events that are not represented in the annotation library, and were assessed by Fisher's exact test and SI change. For this study, only candidates with total junction reads ≥15 and |ΔSI| ≥ 10% with an FDR adjust p-value < 0.2 were selected. Heat maps for cassette events were generated using Cluster and TreeView[70].

**Plasmid constructions**. Minigene sequences were amplified from human placenta genomic DNA (Sigma) using PrimeSTAR Max DNA polymerase (Clontech) and subcloned into pcDNA3.1 + plasmid using BamHI and XhoI restriction sites. Deletions and point mutations were introduced into minigene plasmid by PCR mutagenesis using KAPA HiFi polymerase (KAPA Biosystems) with primers containing respective mutations. Coding sequences of protein were amplified by PrimeSTAR Max DNA polymerase (Clontech) and cloned into pLenti6 vector with respective restriction sites. shRNA sequences were designed using RNAi Platform (Broad Institute) and cloned into pLKO.1_puro plasmid using AgeI and EcoRI restriction sites.

**Transfection**. For minigene analysis, cells were transfected with 0.5 µg minigene plasmid or mixtures of 0.5 µg minigene plasmid and 1.5 µg cDNA expression plasmid, using Lipofectamine 2000 (Invitrogen) at 1:3 ratio (DNA:reagent) for 2 days. For overexpression and knockdown assays, cells were transfected with 2 µg protein overexpression constructs or shRNA plasmids using Lipofectamine 2000 (Invitrogen) for 2 days.

**Splicing analysis by semiquantitative PCR**. RNA was extracted using RNeasy mini kit (Qiagen) and treated with DNaseI on column. A total of 1 µg RNA was used for cDNA synthesis with Advantage RT-for-PCR kit (Clontech). Endogenous cDNA of gene of interest was amplified by FastStart Taq DNA polymerase (Roche) with primers targeting constitutive exons while minigene cDNA was amplified with vector specific primers. PCR cycle condition as follow: initial denaturation at 95 °C for 5 min followed by 35 cycles of 95 °C for 30 s, 60 °C for 30 s, and 72 °C for 30 s. For semiquantification PCR, forward primers were 5′-end radiolabelled using T4 polynucleotide kinase (T4 PNK, New England Biolabs) and γ-$^{32}$P-ATP (Perkin Elmer) followed by purification using illustra Microspin G25 columns (GE Healthcare). PCR condition is same as conventional with a lower cycle number ranging between 21–28 cycles. PCR products were separated by 6% native polyacrylamide gel electrophoresis (PAGE), followed by gel drying. Dried gel was exposed to phosphor-storage screen (GE Healthcare) and signal was scanned using Typhoon imager (GE Healthcare). Band intensities were quantified using ImageQuant TL software (GE Healthcare). PCR products were identified by gel purification and Sanger sequencing. Sequences of primers are listed in Supplementary Table 2.

**In vitro transcription**. DNA template for in vitro transcription was generated by PCR or primer extension with primers containing T7 promoter sequences (5′-CGAAATTAATACGACTCACTATAGGG at forward primer) and sequence of interest. RiboMAX™ Large Scale RNA Production Systems (Promega) was used for in vitro transcription. Newly synthesized RNA was purified by RNeasy mini kit (Qiagen) or mirVana miRNA Isolation Kit (Ambion).

**RNA electrophoretic mobility shift assay**. A total of 50 pmol RNA probe was first dephosphorylated using rSAP (New England Biolabs) and then 5′-end labeled with $^{32}$P using γ-$^{32}$P-ATP (Perkin Elmer) and T4 PNK (New England Biolabs) followed by G25 column purification (GE healthcare). For each REMSA reaction, 0.5 pmol labeled RNA probe was incubated with recombinant ADAR1 or ADAR2 proteins in 1× REMSA binding buffer (Thermo Scientific) supplemented with 0.1 µg/µl yeast tRNA (Invitrogen) at room temperature for 30 min. For competitive binding, 50 pmol unlabelled probe was added to binding mixture. After incubation, binding mixture was separated by 6% native PAGE, followed by gel drying and gel exposure to BioMax® MS film (Carestream Kodak). Sequences of primers are listed in Supplementary Table 2.

**In vitro RNA–protein binding assay by UV crosslinking**. A total of 0.5 pmol $^{32}$P-labeled RNA probes were incubated with recombinant ADAR2 and/or U2AF65 proteins in 1× REMSA binding buffer (Thermo Scientific) supplemented with 0.1 µg/µl yeast tRNA (Invitrogen) at room temperature for 30 min. Samples were then UV irradiated with an energy of $3 \times 10^5$ µJ on ice using Stratalinker UV Crosslinker. Subsequently, samples were mixed with 1× sodium dodecyl sulfate (SDS) loading dye and heat denatured at 95 °C for 5 min followed by SDS–PAGE and gel drying. Protein–RNA bands were visualized by X-ray film development.

**Recombinant protein purification**. HEK293T cells were transfected with FLAG-ADAR1, FLAG-ADAR2, FLAG-U2AF65, or FLAG-SRSF7 overexpression plasmid for 2 days. Cell lysate was collected in lysis buffer (50 mM Tris, pH 7.5, 150 mM NaCl, 1 mM EDTA, 1% TritonX, and cOmplete protease inhibitor (Roche)) and incubated with anti-FLAG M2 magnetic beads (Sigma; one 15-cm dish cell lysate to

15 μl beads) overnight at 4 °C with rotation. Beads was washed with 1× TBS buffer (0.5 M Tris and 1.5 M NaCl) for three times followed by elution in 3× FLAG solution (Sigma). Purified protein was examined by SDS–PAGE and silver staining.

**Nuclear extraction**. Nuclear extraction was performed as per kit manual (Active motif).

**RNA immunoprecipitation**. A 10-cm dish of HEK293T was transfected with 5 μg of wild-type minigene plasmid. At 24 h post transfection, cells were split into two or three 10-cm dishes and transfected with 10 μg of FLAG, FLAG-ADAR1, or FLAG-ADAR2 individually. At 48 h post transfection, cells were collected and lysed in buffer containing 50 mM Tris, pH 7.5, 150 mM NaCl, 1 mM EDTA, and 1% TritonX supplemented with cOmplete protease inhibitor (Roche) and SUPERase In (Invitrogen). Lysate was then incubated with anti-FLAG M2 magnetic beads (Sigma) overnight at 4 °C with rotation followed by six times of washing with 1× TBS buffer (0.5 M Tris and 1.5 M NaCl). A total of 10% of beads was used for protein elution while the rest was subjected to RNA extraction using RNeasy miniprep kit (Qiagen). Extracted RNA was reverse transcribed using Advantage RT-for PCR kit (Clontech) with oligo dT and subsequently qPCR was performed. %Input = $2^{-\Delta Ct} \times 100\%$; $\Delta Ct = Ct_{RIP} - [Ct_{input} - \text{dilution factor}]$. Sequences of primers are listed in Supplementary Table 2.

**RNA pulldown assay**. RNA probe was generated by RiboMAX™ Large Scale RNA Production Systems (Promega) using DNA template containing 5′-T7 promoter, sequence of interest and 3′-aptamer. A total of 25 μg RNA probe was incubated with 50 μl Dynabeads MyOne C1 (Invitrogen) in 300 μl binding buffer (100 mM NaCl, 10 mM MgCl₂, 50 mM Hepes, pH 7.4, and 0.5% Igegal CA-630) for 30 min at 4 °C with rotation followed by three washes with washing buffer (250 mM NaCl, 10 mM MgCl₂, 50 mM Hepes, pH 7.4, and 0.5% Igegal CA-630). For each reaction, 400 μg nuclear extract or 1 mg whole cell extract was diluted in 300 μl washing buffer and supplemented with 4 μl 10 mg/ml yeast tRNA (Invitrogen) and SUPERase In (Invitrogen). RNA immobilized beads were incubated with protein mixtures for 30 min at 4 °C or room temperature with rotation. After three washes, bound proteins were eluted in 2× Laemmli buffer (Sigma) at 95 °C and analyzed by western blot (WB). Sequences of probes are listed in Supplementary Table 2.

**CHX treatment**. HEK293T cells were treated with dimethyl sulfoxide (DMSO) or 100 μg/ml CHX for 4 h followed by RNA extraction and reverse transcription PCR (RT-PCR).

**Western blot**. Protein lysates were prepared with RIPA buffer (Sigma) supplemented with 1× cOmplete EDTA-free protease inhibitor cocktail (Roche) and quantified using Bradford assay (Bio-Rad). Protein lysates were then separated by 8–10% SDS–PAGE followed by incubation with primary antibodies (1:1000 dilution) overnight at 4 °C and incubation with secondary antibodies (1:10,000 dilution) at room temperature for 1 h. Primary antibodies used are anti-ADAR1 (Abcam, ab88574), anti-ADAR2 (Sigma, sab1405426), anti-β actin HRP (Santa Cruz Biotechnology, sc-47778HRP), anti-FLAG HRP (Sigma, A8592), anti-hnRNPK (Santa Cruz Biotechnology, sc-28380), anti-SRSF7 (Abcam, ab138022), anti-U2AF65 (Santa Cruz Biotechnology, sc-53942), and anti-CCDC15 (Abcam, ab170396).

**Foci formation assay**. Cells were seeded to six-well plate at a density of 500 cells/well and incubated at 37 °C for 7–9 days. Cells were stained with crystal violet solution (0.1% crystal violet, 20% methanol in water) for colony visualisation.

**Soft agar assay**. One thousand cells in 0.4% low melted agarose were added on top of solidified 0.6% low melted agarose in each well of six-well plate. Colonies were stained with crystal violet solution (0.05% crystal violet, 20% methanol in water) for visualisation.

**In vivo tumorigenicity assay**. One million cells were injected subcutaneously to the right and left flanks of 4–6-weeks old NOD scid gamma mice. Tumor formation was monitored over a 3-week period and tumor volume was measured every other day once evident tumors were formed. Tumor volume was calculated using the formula: $0.5 \times \text{length} \times \text{width}^2$. All animal experiments were approved by and performed in accordance with the Institutional Animal Care and Use Committees of National University of Singapore (NUS; Singapore).

**Motif analysis**. Motif discovery using MEME[47] was performed to identify any motif enriched in ADARs-regulated cassette exons. Positive input includes 56 high-confidence ADAR1/2-regulated cassette exons (total junction reads ≥15, splicing index change (|ΔSI|) ≥ 10% with an FDR adjust p-value < 0.2; and SI change from at least one knockdown and overexpression experiments must be in opposite direction) with 250 nt flanking sequences at both sides. Control input contains 56 non-ADAR1 and 2-regulated cassette exons (|ΔSI| < 10% in both ADAR1 and ADAR2 overexpression samples) with 250 nt flanking sequences at both sides. The

parameter setting for MEME was any number of repetitions per sequence, a minimum motif width of 6 nt, a maximum motif width of 9 nt, and a search of only the given strand. Input sequences are shown in Supplementary Data 1.

**Statistical analysis**. Two-tailed paired t-test was used for statistical analysis of splicing changes between control and treatment samples (Figs. 2a, c, 3b, d, 4b and 5b, d). Linear regression analysis was used to determine the significance of dose-effect of ADAR overexpression on splicing modulation (Fig. 2b). Unpaired, two-tailed Student's t-test was done for RIP-qPCR, gene expression, in vitro functional assays, and in vivo tumorigenicity assays (Figs. 4f, 5f, 7c and 8c–k, Supplementary Fig. 5e). For all figures: *P < 0.05; **P < 0.01;***P < 0.001; ****P < 0.0001.

**Reporting summary**. Further information on research design is available in the Nature Research Reporting Summary linked to this article.

## Data availability

A reporting summary for this article is available as a Supplementary Information file. The RNA-Seq data are available under accession GSE131658 at Gene Expression Omnibus (GEO). The source data underlying Figs. 1a, d, e, 2a–c, 3b, d, e, 4b, e–g, 5b, d–h, 7b, 8a–k and Supplementary Figs. 1b, 2a, b, 3a, b, f, 4d, f, g, 5a, d–g are provided as a Source Data File. All other data are available from the corresponding author upon reasonable request.

## Code availability

Bioinformatics analysis pipelines are accessible through CSI NGS portal. Bioinformatics codes are available upon request.

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

## Acknowledgements

We thank and acknowledge Prof. Xin-Yuan Guan (The University of Hong Kong, Hong Kong, China) and Dr. Yanru Qin (the first Affiliated Hospital, Zhengzhou University, China) for providing the cDNA samples of ESCC cases. This project was supported by National Research Foundation Singapore; Singapore Ministry of Education under its Research Centres of Excellence initiative; Singapore Ministry of Education's Tier 2 Grants [MOE2018-T2-1-005]; and Singapore Ministry of Education's Tier 3 Grants [MOE2014-T3-1-006].

## Author contributions

L.C. conceived and supervised the study. L.C., S.J.T. and H.S. designed and performed the experiments. H.H. prepared the RNA-seq samples. O.A., J.L. and H.Y. conducted all the bioinformatics analyses. Y.S. assisted in conducting mouse-related experiments. H.H., V.H.E.N., D.J.T.T., Y.S., J.H., F.B.M, K.W.L. and P.P. provided insightful suggestions and experimental materials. S.J.T. and H.S. wrote the manuscript. L.C. edited the manuscript.

## Competing interests

The authors declare no competing interests.
