## [Peer Review File · Nature Communications]

Reviewers' comments:

Reviewer #1 (Remarks to the Author):

This manuscript seeks to dissect ADAR effects on splicing and connect those changes to cancer development. As the author point out, effects of ADAR on splicing have been documented in many systems, but mechanistic details of how ADARs influence splicing are relatively unknown. Herein, the authors provide extensive dissection of the impact of ADAR1 and ADAR2 RNA binding and editing on splicing of two transcripts, CCDC15 and RELL2. In addition, they provide evidence for these two transcripts influencing oncogenic properties of cells. The manuscript needs improved in regard to discussion of the technical details of the experiments performed (for example, it is only mentioned in the methods that the RIP experiment in Figure 4f is done on minigenes and not endogenous transcripts), the results from each experiment and avoidance/explanation of specific splicing terms (ISS).

Points that would support the conclusions in the manuscript are listed below:

1. The splicing analysis of high-throughput sequencing data uses a low cut-off of p-value 0.05. If the authors account for multiple testing in the large dataset and use a p-adjusted cut-off, how many transcripts exhibit splicing changes that are affected by ADAR1 and ADAR2?

2. In nearly every figure, the authors provide semi-quantitative RT-PCR analysis of transcripts. For most, representative images from one replicate are located above the quantitation of multiple replicates. However, in many cases, the representative image is difficult to ascertain changes, for example Figure 2b the effect of ADAR2 on CCDC15 in EC109 cells. For a few targets/experiments (such as those in Figure 2), can the authors provide all three representative images of the RT-PCR experiments and/or comment on the variability? This is particularly important as the final figure connecting splicing changes in Normal and Tumor tissues does not contain quantitation/only representative images.

In addition, for the significance of the semi-quantitative RT-PCR results, it is unclear why the authors used paired t-tests as opposed to ANOVA or other multiple variable tests. While some effects may have minor significance (due to technical repeats as opposed to biological consequences), attention to those small effects detracts from the main conclusions of the paper.

3. The results regarding ADAR2 binding to the GA motif in RELL2 and inhibits U2AF65 binding are not convincing. The RNA binding experiments with ADAR2 and U2AF65 alone as well as together (Figure 5g) seem to suggest that ADAR2 binding to the RELL2 Py3 RNA is enhanced in the presence of U2AF65. Can the authors provide relative affinities (using a titration curve) for ADAR2 binding to this sequence in the presence and absence of U2AF65?

The authors also state that ADAR2 binding to the RELL2 Py3 RNA was diminished by the exonic GA double mutations, but the gel shift provided (Supplemental Figure 4f) shows a large amount of binding. Can the authors provide relative affinities (using a titration curve) for ADAR2 binding to the mutant and wild-type RELL2 Py3 RNA?

4. The impacts of lack of the CCDC15 exon 9 included transcript on soft agar, clonogenic growth and tumor volume are impactful. However, it is unclear whether ADAR1 has a similar function? Or ADAR1 status contributes to the effects of CCDC15 exon 9 included? Similar questions for ADAR2 and RELL2 exon 3 included are also important to address.

Minor points with experiments/results:

1. The western blot analysis of ADAR1 in Figure 1a seems to be out of alignment with the known sizes of ADAR1 p110 and ADAR1 p150.

2. Figure 2 has parallel experiments performed in HEK293T and EC109 cells, except for the dose-dependent response. Is there a reason the authors did not include that cell line? Also, the statistical analysis of the dose-dependent response is lacking.

3. For the MEME analysis (Figure 6), can the authors speculate on what it means that 3 of the 4 most enriched motifs in the ADAR1/2 cassette exons do not contain adenosine?

Minor points with text:

1. There are a number of typographical errors in the discussion (bottom page 13, page 14—omissions of the word “the”) and introduction (bottom of page 3—majority of studies sentence).

2. References 37 and 43 are identical.

Reviewer #2 (Remarks to the Author):

This study addresses the interplay between RNA editing and splicing in the context of cancer. Authors started by modulating the expression of ADAR enzymes in an esophageal squamous carcinoma cell line. Overexpression or silencing ADAR 1 and ADAR2 had global effects on both A-to-I editing and alternative splicing (Fig. 1). Next, authors focused on RNA processing of transcripts from the CCDC15 and RELL2 genes for further analysis. Making use of ADAR mutants devoid of either enzymatic activity or dsRNA-binding capability, the results presented in Fig. 2 elegantly show that both editing-dependent and -independent mechanisms are involved in ADAR-mediated splicing regulation. Potential editing sites were identified in CCDC15 exon 9 and flanking introns and a minigene was mutated to mimic 100% editing at each site; these mutations altered exon 9 inclusion by a mechanism that likely involves binding of SRSF7 to the edited sequence (Fig. 3). Deletion and RNA binding analysis further suggest binding of ADAR enzymes to an intronic dsRNA spanning sequences in intron 8 and intron 9 (Fig.4). In the case of RELL2, ADAR2 binds to a dsRNA that forms at the Py-tract, thereby competing for the binding of U2AF65, which is essential for spliceosome assembly (Fig. 5). Based on the results obtained with CCDC15 and RELL2 reporter genes, authors apply bioinformatics tools to predict ADAR-binding motifs involved in ADAR-mediated splicing regulation (Fig. 6). Finally, the potential biological consequences of ADAR-mediated splicing regulation are investigated, particularly in the context of cancer (Figs. 7 and 8).

Overall, this is a timely and sound study; the manuscript is very well presented, and conclusions are supported by convincing experimental results. I have no criticisms. I have, however, some suggestions that authors may find useful to address on a revised version.

1. A global study of ADAR1-RNA interaction in human cells using CLIP-seq has been reported (Bahn et al Nat Comm 2015). Can authors use these datasets to validate the predicted binding to GA-rich motifs?

2. A Western blot showing downregulation of RELL2 protein would unequivocally demonstrate the physiological impact of exon 3 skipping.

3. A recent study proposed that RELL2 acts as a tumor suppressor (Wang et al Eur J Pharmacol 2019). This work should be cited and discussed, as it re-enforces the findings reported here.

Reviewer #3 (Remarks to the Author):

In this manuscript, Tang et al. investigated the global effects of ADAR1/2 on RNA splicing, and

discovered more than one hundred high-confident splicing events regulated by ADAR1 and/or ADAR2. They then chose two affected splicing events in CCDC15 and RELL2 as examples for further mechanistic and functional studies. The authors revealed that ADAR1/2 tend to bind to GA-rich dsRNAs for RNA editing and splicing regulation. In particular, their bindings to dsRNAs affect the recruitment of splicing factors such as SRSF7 and U2AF65, leading to repression of exon inclusion. Finally, they also showed that ADAR-affected splicing events in CCDC15 and RELL2 have significant impact on in vitro colony-formation of tumor cells and also in vivo tumorigenesis. Overall, this is an interesting and timely study, which provide compelling evidence showing that ADAR affect RNA splicing through their RNA editing dependent and independent activities, which in turn affect tumorigenesis. This manuscript is well written and ease to understand.

Nonetheless, I have a few questions/comments below that should be addressed by the authors:

1) There are discrepancies in the number of bands and sizes of ADAR1 in WB data. Figure 1A shows two bands for ADAR1, with one band \sim 130kD and another below 100kD. However, there is a single band for ADAR1 at \sim 130kD in Figure S2A and a single band at 100kD in Figure S2B. Could this be due to cell type differences and why? Similarly, ADAR2 overexpression sometimes produces one band (Fig 1A and Fig S2A for EC109) or two bands (Fig S2A 293T, Fig S2B). The authors should also provide uncropped WB's in the supplementary figures.

2) In Figure 3D, the authors showed that mutation of site 2 from A to G facilitates exon 9 exclusion, and even more so when SRSF7 is overexpressed. This implied that SRSF7 binding is affected by ADAR1 editing. Can the authors validate whether RNA binding of ADAR1/2 affects SRSF7 binding? Do the mutations introduced in the intron 9 region (Figure 4), particularly del5, affect SRSF7 binding to RNA?

3) Since both ADAR1 and ADAR2 influence CCDC15 splicing in the same direction, can the effects of ADAR1 depletion be compensated by ADAR2 and vice versa? Does double knockdown of both ADARs have an additive effect?

4) From the text, 23 ADAR1/2 high and 10 ADAR1/2 normal/low patient samples were tested. However in Figures 8a and b, data is presented from only 6 samples each for ADAR1/2 high and ADAR1/2 normal/low patient samples. An additional graph summarizing the data from all the patient samples would be helpful (e.g. a bar graph of PSI of CCDC15 and RELL2 comparing ADAR1/2 high and normal/low groups). This way, statistical analyses comparing the PSI of different patient groups can be done, although the small sample size might limit this type of analysis.

5) WB showing CCDC15 exon9 knockdown should be done to show that the effect of knockdown is specific for that particular isoform at the protein level, and that other isoforms are unaffected. In addition, these experiments (foci, soft agar, mouse experiments) should also include overexpression of CCDC15 containing exon9.

6) Although unrelated to the main focus of the manuscript, it would be interesting to discuss what makes CCDC15 exon 9 isoform oncogenic in esophageal cancer. A brief discussion in the Discussion section would be helpful.

7) Addition of a schematic summarizing the findings in the manuscript can be added to help the reader.

Editorial Board:

Your manuscript entitled "Cis- and trans-regulation of pre-mRNA splicing by RNA editing enzymes: influencing cancer development" has now been seen by 3 referees. You will see from their comments below that while they find your work of interest, some important points are raised. We are interested in the possibility of publishing your study in Nature Communications, but would like to consider your response to these concerns in the form of a revised manuscript before we make a final decision on publication. We consider it particularly important that the revised manuscript provide all three images of the RT-PCR experiments (Figs. 2-7) and uncropped western blots (Fig. 1, Supplementary fig 2) in a source data file.

- We have included all required images and uncropped blots in the revised source data file. Please also see our point-by-point response to the reviewers' comments and suggestions below.

Reviewer #1:

This manuscript seeks to dissect ADAR effects on splicing and connect those changes to cancer development. As the author point out, effects of ADAR on splicing have been documented in many systems, but mechanistic details of how ADARs influence splicing are relatively unknown. Herein, the authors provide extensive dissection of the impact of ADAR1 and ADAR2 RNA binding and editing on splicing of two transcripts, CCDC15 and RELL2. In addition, they provide evidence for these two transcripts influencing oncogenic properties of cells. The manuscript needs improved in regard to discussion of the technical details of the experiments performed (for example, it is only mentioned in the methods that the RIP experiment in Figure 4f is done on minigenes and not endogenous transcripts), the results from each experiment and avoidance/explanation of specific splicing terms (ISS).

-We thank the reviewer for pointing this out. We have carefully checked through all the figure legends and provided more details for the experiments performed and added explanations of ISS.

Points that would support the conclusions in the manuscript are listed below:

1. *The splicing analysis of high-throughput sequencing data uses a low cut-off of p-value 0.05. If the authors account for multiple testing in the large dataset and use a p-adjusted cut-off, how many transcripts exhibit splicing changes that are affected by ADAR1 and ADAR2?*

-Thank the reviewer for raising this question. We reanalysed our data by setting a FDR-adjusted p-value cut-off of 0.05, 0.1 or 0.2 and found that the number of ADAR1/2-regulated splicing events dropped (please see the tables below). For the following two reasons, we decided to set an adjust p-value cut-off of 0.2 (FDR<0.2) and amended main text and **Fig.1b-c** accordingly.

- 1) If we set FDR<0.1 or FDR<0.05, most of our validated cassette exon events was filtered out, such as *CCDC15* exon 9, *ZNF517* exon 3, *RELL2* exon 3 and *AKAP9* exon 19.
- 2) As discussed in Handbook of Biological Statistics¹, low FDR cut-off may lead to high rate of false negative.

We hope to convince the reviewer that in our study, we have set stringent requirements, including 1) total junction reads ≥ 15 , 2) splicing index change ($|\Delta SI|$) $\geq 10\%$ with a FDR value < 0.2 ; and 3) particularly SI change from both knockdown and overexpression experiments must be in opposite direction. All six randomly selected targets could be experimentally validated in both EC109 cells and HEK293T cells (**Fig. 1d,e** and **Supplementary Fig. 1b**).

Type of splicing	No. of ADAR1-regulated events			
	P<0.05	Adjusted p-value		
		FDR<0.2	FDR<0.1	FDR<0.05
Cassette exon	19	19	10	6
Intron retention	22	16	9	3
Alternative splice-site	14	6	1	0

Type of splicing	No. of ADAR2-regulated events			
	P<0.05	Adjusted p-value		
		FDR<0.2	FDR<0.1	FDR<0.05
Cassette exon	28	21	12	7
Intron retention	14	12	4	2
Alternative splice-site	24	10	2	0

2. In nearly every figure, the authors provide semi-quantitative RT-PCR analysis of transcripts. For most, representative images from one replicate are located above the quantitation of multiple replicates. However, in many cases, the representative image is difficult to ascertain changes, for example Figure 2b the effect of ADAR2 on CCDC15 in EC109 cells. For a few targets/experiments (such as those in Figure 2), can the authors provide all three representative images of the RT-PCR experiments and/or comment on the variability? This is particularly important as the final figure connecting splicing changes in Normal and Tumor tissues does not contain quantitation/only representative images.

In addition, for the significance of the semi-quantitative RT-PCR results, it is unclear why the authors used paired t-tests as opposed to ANOVA or other multiple variable tests. While some effects may have minor significance (due to technical repeats as opposed to biological consequences), attention to those small effects detracts from the main conclusions of the paper.

- As requested by the reviewer, all images of RT-PCR experiments (biological replicates) are provided in **Source data file**.

-The reviewer has commented “in many cases, the representative image is difficult to ascertain changes”. In this study, we did observe small batch-to-batch variations in the baseline PSI values of EV or Scr control cells, which was most likely due to the difference in cell conditions arising from different passage number, cell confluency or transfection efficiency and so on. That was also why for each batch of RT-PCR experiments, we included at least 3 biological replicates, in order to make solid conclusions.

- The reasons why we used the paired t-test include:

- 1) we intended to compare each treatment to its own empty vector (EV) control (e.g WT vs EV, DeAD vs EV, and EAA vs EV) for each biological replicate;
- 2) as abovementioned, the baseline PSI values of EV control cells may be varied; and
- 3) ANOVA and other multiple variable tests are used to compare more than two groups and show significance as long as one of the comparing groups shows the difference.

Therefore, as we intended to show whether there was a repressive effect of WT, DeAD or EAA on exon inclusion, we think the paired t-test is more suitable than ANOVA.

3. The results regarding ADAR2 binding to the GA motif in RELL2 and inhibits U2AF65 binding are not convincing. The RNA binding experiments with ADAR2 and U2AF65 alone as well as together (Figure 5g) seem to suggest that ADAR2 binding to the RELL2 Py3 RNA is enhanced in

the presence of U2AF65. Can the authors provide relative affinities (using a titration curve) for ADAR2 binding to this sequence in the presence and absence of U2AF65?

-We thank the reviewer for making valuable comments. We first repeated the original UV-crosslink experiment twice and confirmed there was no obvious enhancement of ADAR2 binding to *RELL2* in the presence of U2AF65 (**revised Fig. 5g**). Next, we conducted the UV-crosslink assay with increasing amount of U2AF65 protein and we also did not observe a dose-dependent increase in ADAR2 binding to *RELL2* (**Supplementary Fig. 4h**).

The authors also state that ADAR2 binding to the RELL2 Py3 RNA was diminished by the exonic GA double mutations, but the gel shift provided (Supplemental Figure 4f) shows a large amount of binding. Can the authors provide relative affinities (using a titration curve) for ADAR2 binding to the mutant and wild-type RELL2 Py3 RNA?

-Thank the reviewer for pointing this out. We performed REMSA with increasing amount of ADAR2 protein for both wild-type and mutant *RELL2* probes (**revised Fig. 5e**). Our new REMSA data showed that ADAR2 protein binds more strongly to the wild-type probe than the mutant probe by measuring the intensity of RNA-protein shift band over the total intensity (sum of the intensities of free probe and shift band) (**revised Fig. 5e**). The REMSA data suggests that binding of ADAR2 to *RELL2* was reduced by double mutations. We have replaced the original Fig. 5e with this new data including both gel image and band intensity measurements.

4. The impacts of lack of the CCDC15 exon 9 included transcript on soft agar, clonogenic growth and tumor volume are impactful. However, it is unclear whether ADAR1 has a similar function? Or ADAR1 status contributes to the effects of CCDC15 exon 9 included? Similar questions for ADAR2 and RELL2 exon 3 included are also important to address.

- We thank the reviewer for raising these questions. To further confirm the functional impacts of *CCDC15* exon 9 inclusion, we have included new data showing the functional role of *CCDC15* exon 9-included (full-length) isoform in **Fig. 8f-h**. Consistently, overexpression of *CCDC15* exon 9-included isoform enhanced the tumorigenicity of EC109 cells.

- Functions of ADAR1 and ADAR2 in ESCC (esophageal squamous cell carcinoma) have been extensively studied previously^{2,3}. ADAR1 generally acts as an oncogene in ESCC progression; while ADAR2 acts as a tumor suppressor. In this study, our main focus was to investigate whether and how ADAR1 and ADAR2 regulate cancer-associated splicing events, rather than study if ADARs are the main contributors to tumorigenic or tumor suppressive effect of *CCDC15*-ex9-included and *RELL2*-ex3-included isoforms. Moreover, ADAR1 and ADAR2 are involved in many cellular processes such as protein recoding and miRNA biogenesis. Modulation of ADAR1 or ADAR2 expression level would affect multiple downstream targets. In addition, as we discussed in the manuscript, splicing of *CCDC15* and *RELL2* can be regulated by other splicing factors and their splicing patterns in tumors may be a combinational outcome of different splicing regulators.

- In this study, we showed that both ADAR1 and ADAR2 repress *CCDC15* exon 9 inclusion. As shown in **Supplementary Fig. 5b**, oncogenic *CCDC15*-ex9-included isoform was upregulated in 60% (20/33) of primary ESCC tumor samples, suggesting that loss of ADAR2 may contribute more to the increased *CCDC15* exon 9 inclusion in tumors than overexpression of ADAR1. Therefore, we did not perform experiments to study whether that ADAR1 status contributes to the oncogenic effect of *CCDC15* exon 9 included isoform.

-For *RELL2*, we found that ADAR2 represses *RELL2* exon 3 inclusion in EC109 cells. Although ADAR2 is thought to be a tumor suppressor and downregulated in ESCC, we did observe that 30% (10/33) of ESCC patients showed higher expression of ADAR2 in tumors (**Supplementary Fig. 5b**). For this group of patients, ADAR2 may contribute to oncogenic effect of *RELL2* ex3 exclusion.

For the remaining ESCC patients, there may be other splicing factors contributing to the tumor suppressive effect of *RELL2* ex3 exclusion, which remains for our further investigation.

Minor points with experiments/results:

1. The western blot analysis of *ADAR1* in Figure 1a seems to be out of alignment with the known sizes of *ADAR1* p110 and *ADAR1* p150.

- Thank you for pointing out the misalignment. We have corrected the marker labelling.

2. Figure 2 has parallel experiments performed in HEK293T and EC109 cells, except for the dose-dependent response. Is there a reason the authors did not include that cell line? Also, the statistical analysis of the dose-dependent response is lacking.

- We have applied the linear regression analysis to show that ADAR proteins could cause significant *CCDC15* splicing change in HEK293T dose-dependently (**Fig. 2b**).

- We were facing difficulties in obtaining a good dose-dependent overexpression of ADAR1 and ADAR2 proteins in EC109 cells, we thus observed a saturated repressive effect on *CCDC15* exon inclusion upon ADAR1/2 overexpression (please see the figure below). Therefore, we did not include these data in the manuscript.

a, RT-PCR analysis of *CCDC15* exon 9 splicing in EC109 cells overexpressed with dose-increase ADAR1 or ADAR2. b, Western blot analysis of ADAR1 and ADAR2 expression in EC109 cells.

3. For the MEME analysis (Figure 6), can the authors speculate on what it means that 3 of the 4 most enriched motifs in the ADAR1/2 cassette exons do not contain adenosine?

-We thank the reviewer for raising this question. The frequency of adenosine in our input sequences is about 25% and therefore lack of adenosine enrichment within the four discovered motifs is not due to the depletion of adenosine in our inputs. Adenosine is not completely absent from the other three motif but rather it is less frequent and conserved in these three motifs. One possibility for this observation could be that these cassette exons are regulated by splicing factors that recognize pyrimidine-rich sequences and therefore we could observe relatively pyrimidine-rich motifs in our analysis. Another possibility is that adenosine is relatively less conserved in the length of motif we set because we could see frequent conserved adenosine nucleotide in longer motif sequences.

Minor points with text:

1. *There are a number of typographical errors in the discussion (bottom page 13, page 14—omissions of the word “the”) and introduction (bottom of page 3—majority of studies sentence).*

- Thank the reviewer for pointing this out. We have corrected the text as highlighted in red front.

2. *References 37 and 43 are identical.*

- Thank the reviewer for pointing this out. We have removed the redundant references.

Reviewer #2:

This study addresses the interplay between RNA editing and splicing in the context of cancer. Authors started by modulating the expression of ADAR enzymes in an esophageal squamous carcinoma cell line. Overexpression or silencing ADAR 1 and ADAR2 had global effects on both A-to-I editing and alternative splicing (Fig. 1). Next, authors focused on RNA processing of transcripts from the CCDC15 and RELL2 genes for further analysis. Making use of ADAR mutants devoid of either enzymatic activity or dsRNA-binding capability, the results presented in Fig. 2 elegantly show that both editing-dependent and –independent mechanisms are involved in ADAR-mediated splicing regulation. Potential editing sites were identified in CCDC15 exon 9 and flanking introns and a minigene was mutated to mimic 100% editing at each site; these mutations altered exon 9 inclusion by a mechanism that likely involves binding of SRSF7 to the edited sequence (Fig. 3). Deletion and RNA binding analysis further suggest binding of ADAR enzymes to an intronic dsRNA spanning sequences in intron 8 and intron 9 (Fig.4). In the case of RELL2, ADAR2 binds to a dsRNA that forms at the Py-tract, thereby competing for the binding of U2AF65, which is essential for spliceosome assembly (Fig. 5). Based on the results obtained with CCDC15 and RELL2 reporter genes, authors apply bioinformatics tools to predict ADAR-binding motifs involved in ADAR-mediated splicing regulation (Fig. 6). Finally, the potential biological consequences of ADAR-mediated splicing regulation are investigated, particularly in the context of cancer (Figs. 7 and 8).

Overall, this is a timely and sound study; the manuscript is very well presented, and conclusions are supported by convincing experimental results. I have no criticisms. I have, however, some suggestions that authors may find useful to address on a revised version.

1. A global study of ADAR1-RNA interaction in human cells using CLIP-seq has been reported (Bahn et al Nat Comm 2015). Can authors use these datasets to validate the predicted binding to GA-rich motifs?

-We thank the reviewer for giving this valuable suggestion. Bahn et al. (2015) reported a 50nt-long conserved ADAR1 binding motif which is located at consensus *Alu* elements. In our study, we observed a short GA-rich sequence (indicated in blue dotted box) towards the 3' end of their reported motif. Next, we checked for individual splicing targets regulated by ADAR1 and found that about 25 % of our identified splicing targets have binding peaks at the flanking introns, despite the location of peaks are distal from exon-intron junction. Nonetheless, our GA-rich sequence should be enriched in targets which undergo ADAR1-regulated splicing change; however the ADAR1 CLIP-Seq conducted by Bahn et al. captured all ADAR1 binding sites in a glioma cell line U87MG without particular interest in target transcripts which may undergo splicing changes. Therefore, by looking at the ADAR1 CLIP-Seq data may not be suitable to validate our GA-rich motif.

2. A Western blot showing downregulation of *RELL2* protein would unequivocally demonstrate the physiological impact of exon 3 skipping.

-As suggested by the reviewer, we have tried different antibodies against *RELL2* including the one used in a previous publication (Wang et al. Eur J Pharmacol 2019) for western blot analysis, but we could not obtain a good WB result. Due to the poor antibody quality and low endogenous expression of *RELL2* in EC109 cells, we could only detect a weak *RELL2* band with high background signals. As shown in the figure below, there was a subtle decrease in *RELL2* protein upon ADAR2 overexpression. In addition, ADAR2 overexpression could only repress *RELL2* exon 3 inclusion level by 20% at maximum (**Fig. 2c**). Therefore, it is challenging to observe a pronounced reduction in *RELL2* protein expression by WB analysis. Due to the poor quality of data, we decided not to include this data in the manuscript. Nonetheless, we could include the images in the paper if requested.

Western blot analysis of *RELL2* expression.

Lane 1: Cell lysate from EC109 cells overexpressed with empty vector.

Lane 2: Cell lysate from EC109 cells overexpressed with *ADAR2* expression construct.

Lane 3: Cell lysate from EC109 cells overexpressed with *RELL2* expression construct

Red arrow indicates the *RELL2* protein band.

3. A recent study proposed that *RELL2* acts as a tumor suppressor (Wang et al Eur J Pharmacol 2019). This work should be cited and discussed, as it re-enforces the findings reported here.

- Thank you for your suggestion. We have added a short discussion into the revised manuscript.

Reviewer #3:

In this manuscript, Tang et al. investigated the global effects of ADAR1/2 on RNA splicing, and discovered more than one hundred high-confident splicing events regulated by ADAR1 and/or ADAR2. They then chose two affected splicing events in CCDC15 and RELL2 as examples for further mechanistic and functional studies. The authors revealed that ADAR1/2 tend to bind to GA-rich dsRNAs for RNA editing and splicing regulation. In particular, their bindings to dsRNAs affect the recruitment of splicing factors such as SRSF7 and U2AF65, leading to repression of exon inclusion. Finally, they also showed that ADAR-affected splicing events in CCDC15 and RELL2 have significant impact on in vitro colony-formation of tumor cells and also in vivo tumorigenesis. Overall, this is an interesting and timely study, which provide compelling evidence showing that ADAR affect RNA splicing through their RNA editing dependent and independent activities, which in turn affect tumorigenesis. This manuscript is well written and ease to understand.

Nonetheless, I have a few questions/comments below that should be addressed by the authors:

1) *There are discrepancies in the number of bands and sizes of ADAR1 in WB data. Figure 1A shows two bands for ADAR1, with one band ~130kD and another below 100kD. However, there is a single band for ADAR1 at ~130kD in Figure S2A and a single band at 100kD in Figure S2B. Could this be due to cell type differences and why? Similarly, ADAR2 overexpression sometimes produces one band (Fig 1A and Fig S2A for EC109) or two bands (Fig S2A 293T, Fig S2B). The authors should also provide uncropped WB's in the supplementary figures.*

- We thank the reviewer for raising this question and apologize that the discrepancies of p110 band among Fig. 1a, Supplementary Fig. 2a and 2b were due to misalignment when we labeled the images and we have corrected the marker labeling.

- There are two ADAR1 isoforms, p110 and p150, and the major isoform is the p110 isoform at ~110kD. The p150 band is not always visible as its expression is induced by interferon^{4,5} and therefore can be affected by cell conditions. Similarly, ADAR2 has different isoforms, mainly due to alternative splicing^{6,7}, and different ADAR2 isoforms can be differentially expressed under different cell conditions. In addition, when we prolonged the exposure time for western blot, we were able to observe two bands of ADAR2.

- As requested by the reviewer, we have provided uncropped WB images in the **Source Data file**.

2) *In Figure 3D, the authors showed that mutation of site 2 from A to G facilitates exon 9 exclusion, and even more so when SRSF7 is overexpressed. This implied that SRSF7 binding is affected by ADAR1 editing. Can the authors validate whether RNA binding of ADAR1/2 affects SRSF7 binding? Do the mutations introduced in the intron 9 region (Figure 4), particularly del5, affect SRSF7 binding to RNA?*

- Thank the reviewer for making this valuable suggestion. We have performed REMSA to show that binding of ADAR1 to the CCDC15 In8-9 dsRNA probe did not affect binding of SRSF7 and included this new data as **Fig. 4g**. In addition, we have conducted RIP assay to check if SRSF7 binds to the mutant del5 minigene, in order to address the reviewer's second question. We could observe a similar binding affinity of SRSF7 to both wild-type and In9 del5 mutant ADAR1 (please see the figure below). We feel this new REMSA data is sufficient to support that ADAR1 binding does not affect the binding of SRSF7 to CCDC15 transcript, so we only included the REMSA data into the revised manuscript. Nonetheless, we could include the RIP data into the manuscript if requested.

HEK293T cells were transfected with FLAG empty vector or FLAG-SRSF7 expression construct together with wild type or In9 del5 mutant *CCDC15* minigene followed by RNA immunoprecipitation. FLAG-SRSF7 could interact with wild type or mutant *CCDC15* minigene transcripts with similar affinity *in vivo*.

3) Since both *ADAR1* and *ADAR2* influence *CCDC15* splicing in the same direction, can the effects of *ADAR1* depletion be compensated by *ADAR2* and vice versa? Does double knockdown of both *ADARs* have an additive effect?

-As suggested by the reviewer, we have tried double knockdown of *ADAR1* and *ADAR2* in HEK293T cells and we could not observe any additive effect on *CCDC15* splicing (please see the figure below). It is likely that due to the relatively low endogenous levels of *ADAR1* and *ADAR2* in HEK293T cells, the inclusion level of *CCDC15* exon 9 is relatively high endogenously. Therefore, further depletion of both *ADAR1* and *ADAR2* could not further enhance the exon 9 inclusion as compared to single knockdown.

a, RT-PCR analysis of *CCDC15* exon 9 splicing in HEK293T cells transfected with *ADAR1* or/and *ADAR2* shRNAs. b, QPCR analysis of *ADAR1* and *ADAR2* expressions in indicated cells.

4) From the text, 23 *ADAR1/2* high and 10 *ADAR1/2* normal/low patient samples were tested. However in Figures 8a and b, data is presented from only 6 samples each for *ADAR1/2* high and *ADAR1/2* normal/low patient samples. An additional graph summarizing the data from all the

patient samples would be helpful (e.g. a bar graph of PSI of CCDC15 and RELL2 comparing ADAR1/2 high and normal/low groups). This way, statistical analyses comparing the PSI of different patient groups can be done, although the small sample size might limit this type of analysis.

- As suggested by the reviewer, we have provided new figure showing the Δ PSI of *CCDC15* and *RELL2* cassette exons for all patient samples (**Supplementary Fig. 5b, c**). However, due to the small sample size, there was no statistical significance between different groups by using Student's *t*-test.

5) *WB showing CCDC15 exon9 knockdown should be done to show that the effect of knockdown is specific for that particular isoform at the protein level, and that other isoforms are unaffected. In addition, these experiments (foci, soft agar, mouse experiments) should also include overexpression of CCDC15 containing exon9.*

- As suggested by the reviewer, we have provided a WB image showing the knockdown of *CCDC15* at protein level (**Supplementary Fig. 5f**). Although we could only observe one single band at the predicted molecular weight, the *CCDC15* antibody seemed not to work very well to detect endogenous *CCDC15* protein (**Supplementary Fig. 5f**).

- There is only 4.85kD difference in the protein size between two isoforms (exon 9-included and -skipped), it is not likely to differentiate them by WB. Therefore, we have used qRT-PCR to show the expressions of two isoforms in *CCDC15* exon 9 knockdown cells in **Supplementary Fig. 5d**.

- We also included the functional role of *CCDC15* exon9-included isoform. As shown in **Fig. 8f-h**, both *in vitro* and *in vivo* tumorigenicity assays supported that overexpression of *CCDC15* exon9-included isoform in EC109 cells promotes tumorigenesis.

6) *Although unrelated to the main focus of the manuscript, it would be interesting to discuss what makes CCDC15 exon 9 isoform oncogenic in esophageal cancer. A brief discussion in the Discussion section would be helpful.*

- Thank the reviewer for the suggestion. We have added a short discussion regarding the possible functional disruption by exon 9 skipping in the "Discussion" (in red font).

7) *Addition of a schematic summarizing the findings in the manuscript can be added to help the reader.*

- Thank the reviewer for the suggestion. We have provided a schematic diagram summarizing our findings in **Fig. 9**.

References

1. McDonald JH. Handbook of biological statistics. (2014).
2. Qin YR, *et al.* Adenosine-to-inosine RNA editing mediated by ADARs in esophageal squamous cell carcinoma. *Cancer Res* **74**, 840-851 (2014).
3. Chen YB, *et al.* ADAR2 functions as a tumor suppressor via editing IGFBP7 in esophageal squamous cell carcinoma. *Int J Oncol* **50**, 622-630 (2017).

4. George CX, Samuel CE. Human RNA-specific adenosine deaminase ADAR1 transcripts possess alternative exon 1 structures that initiate from different promoters, one constitutively active and the other interferon inducible. *Proc Natl Acad Sci U S A* **96**, 4621-4626 (1999).
5. Patterson JB, Samuel CE. Expression and regulation by interferon of a double-stranded-RNA-specific adenosine deaminase from human cells: evidence for two forms of the deaminase. *Mol Cell Biol* **15**, 5376-5388 (1995).
6. Fu Y, Zhao X, Li Z, Wei J, Tian Y. Splicing variants of ADAR2 and ADAR2-mediated RNA editing in glioma. *Oncol Lett* **12**, 788-792 (2016).
7. Li Z, *et al.* Aberrant alternative splicing pattern of ADAR2 downregulates adenosine-to-inosine editing in glioma. *Oncol Rep* **33**, 2845-2852 (2015).

REVIEWERS' COMMENTS:

Reviewer #1 (Remarks to the Author):

The authors have addressed all of my concerns.

Reviewer #2 (Remarks to the Author):

Authors have satisfactorily addressed my comments and the revised manuscript is significantly improved.

Reviewer #3 (Remarks to the Author):

In this revised manuscript, the authors have added additional data, and have carefully and thoroughly addressed all the critiques from the reviewers. As a result, the quality of this paper has been further improved. Overall, this is an interesting and timely study. Therefore, I fully support the publication of this paper in Nature Communications.